JCB Journal of Cell Biology

# REPORT

# StableMARK-decorated microtubules in cells have expanded lattices

Leanne de Jager[1], Klara I. Jansen[2], Robin Hoogebeen[2], Anna Akhmanova[2], Lukas C. Kapitein[2], Friedrich Förster[1], and Stuart C. Howes[1]

**Microtubules are crucial in cells and are regulated by various mechanisms like posttranslational modifications, microtubule-associated proteins, and tubulin isoforms. Recently, the conformation of the microtubule lattice has also emerged as a potential regulatory factor, but it has remained unclear to what extent different lattices co-exist within the cell. Using cryo-electron tomography, we find that, while most microtubules have a compacted lattice (~41 Å monomer spacing), approximately a quarter of the microtubules displayed more expanded lattice spacings. The addition of the microtubule-stabilizing agent Taxol increased the lattice spacing of all microtubules, consistent with results on reconstituted microtubules. Furthermore, correlative cryo-light and electron microscopy revealed that the stable subset of microtubules labeled by StableMARK, a marker for stable microtubules, predominantly displayed a more expanded lattice spacing (~41.9 Å), further suggesting a close connection between lattice expansion and microtubule stability. The coexistence of different lattices and their correlation with stability implicate lattice spacing as an important factor in establishing specific microtubule subsets.**

## Introduction

Microtubules are highly dynamic, polarized cytoskeletal structures that are involved in many diverse functions within cells. These include supporting cell shape by providing mechanical stability and acting as tracks along which motor proteins transport cargo. To efficiently participate in this wide range of functions, microtubules are subject to multiple layers of regulation and modifications that mark different populations of the microtubules and influence the downstream behavior of microtubule-associated proteins (MAPs) and motor proteins. These regulatory components, like tubulin isoforms and posttranslational modifications (PTMs), are termed the tubulin code (Verhey and Gaertig, 2007). The tubulin code is a concept that describes how cells create specialized microtubule subpopulations and control their microtubule networks (Gadadhar et al., 2017; Janke and Magiera, 2020; Roll-Mecak, 2020). However, the link between PTMs and the behaviors and dynamics of different MAPs and motor proteins observed in cells has been difficult to reproduce in in vitro reconstitution experiments (Bär et al., 2022; Burute and Kapitein, 2019), suggesting that additional layers of regulation might exist.

One such additional layer of regulation might come from the structure of the microtubule itself as several studies have shown that proteins are sensitive to its lattice spacing and/or nucleotide state (Akhmanova and Kapitein, 2022; Iwanski and Kapitein, 2023; Manka and Moores, 2018a; Zhang et al., 2018). Upon polymerization of free αβ-tubulin dimers into microtubules, the GTP bound to β-tubulin is hydrolyzed to GDP, and the structural conformation of the αβ-tubulin dimer changes (Alushin et al., 2014). This GDP-bound state, which makes up the bulk of the microtubule shaft, generally is characterized by a compacted average monomer spacing of ~41 Å that is distinct from that of the presumed GTP-bound state at the tip (the GTP-cap). The GTP-bound state is thought to have an expanded lattice spacing of ~42 Å as determined for hydrolysis-deficient mutants and the slowly hydrolysable GTP analogue guanylyl-(α,β)-methylene-diphosphonate (GMPCPP) (Hyman et al., 1995; LaFrance et al., 2022). End-binding (EB) proteins preferentially bind to the GTP cap of the microtubule (Bechstedt and Brouhard, 2012; Tirnauer and Bierer, 2000). However, at saturating concentrations, EB compacts the lattice, even overriding the expanded state established by GMPCPP (Zhang et al., 2018), which further underscores that the lattice itself is malleable. Furthermore, various microtubule-binding proteins have higher affinities for expanded or compacted lattices in vitro and can also induce lattice expansion or compaction, respectively (Ferro et al., 2022; Guedes-Dias et al., 2019; Peet et al., 2018; Shima et al., 2018; Watanabe et al., 2020; Zhang et al., 2017) (Table S1 and Fig. S1). This behavior highlights the plasticity of the lattice and suggests that it might integrate or discriminate between effects

[1]Structural Biochemistry, Department of Chemistry, Bijvoet Centre for Biomolecular Research, Utrecht University, Utrecht, Netherlands; [2]Cell Biology, Neurobiology and Biophysics, Department of Biology, Faculty of Science, Utrecht University, Utrecht, Netherlands.

Correspondence to Stuart C. Howes: s.c.howes@uu.nl; Friedrich Förster: f.g.forster@uu.nl; Lukas C. Kapitein: l.kapitein@uu.nl.



from opposing factors. Nonetheless, whether microtubules in cells also exhibit different lattice spacings has remained unresolved.

Besides lattice changes linked to the nucleotide state and EB proteins, drugs like the anti-cancer drug Taxol (paclitaxel) can alter the lattice spacing of the microtubule in vitro (Kellogg et al., 2017). While this microtubule-stabilizing drug has a long history in the clinical setting, with known side effects, primarily neuropathies (Shin, 2023), the exact mechanism of action and how cancers develop resistance remain unclear (Barbuti and Chen, 2015). The mechanism might rely on disrupted signaling pathways and other secondary alterations that impact cellular health, potentially caused by changes in lattice spacing. These changes are known to alter the binding of MAPs in vitro. Tau and MAP2 envelop formation compacts the microtubule lattice, competing with the expansion supported by Taxol (Siahaan et al., 2022). Similarly, doublecortin, which binds the same site as EB proteins, is excluded from Taxol-expanded regions of the microtubule lattice (Ettinger et al., 2016). This suggests that changes in microtubule lattice spacing may be as important for understanding the mechanism of Taxol action as its stabilizing effect on microtubules. Hence, resolving the effect of Taxol on the microtubule lattice inside cells can help clarify if alterations in lattice spacing could be an additional explanation for Taxol's potent effect on cancer cells. Taken together, variability in lattice spacing has been observed extensively in vitro and can greatly affect protein behavior. However, the microtubule lattice spacing in situ has never been measured directly and its potential to act as a layer of regulation within cells therefore remains uncertain.

## Results and discussion

To examine the lattice spacing of microtubules in cells, we cultured U2OS cells on electron microscopy grids, vitrified them, and thinned the cells, creating ~150-nm-thick lamellae using cryo-focused ion beam (FIB) milling. Subsequently, the lamellae were imaged using cryo-electron tomography (cryo-ET) (Rigort et al., 2012). To measure the lattice spacing of cellular microtubules, we used a layer line approach based on Fourier analysis of microtubule segments (Fig. 1). As tubulin is a repeating unit within the microtubule, its lattice spacing will appear as a strong frequency in Fourier space. Building from typical 2D analysis of in vitro microtubules (Mandelkow et al., 1977), the protocol masks out microtubule density from surrounding cellular material and aligns the segments in 3D prior to projection and computation of the 2D Fourier transforms (Fig. S2 A). Summed power spectra of the aligned microtubule segments were then used to measure the lattice spacing based on maxima in a line profile plot (Fig. 1, B–D). A box size of 1,030 pixels (224-nm edge length) was used for this analysis as it was computationally more efficient than bigger box sizes while having a comparable signal-to-noise ratio (Fig. S2 B). To confirm that our imaging conditions and processing approach were reliable, in vitro microtubule preparations of dynamic-, Taxol-, and GMPCPP-bound microtubules were examined as a technical control to confirm our measurements agreed with literature values (Fig.

S3, A–E). Our analysis recovered the expected lattice spacings, namely predominantly compacted for dynamic microtubules, and expanded for Taxol- and GMPCPP-bound microtubules, verifying our approach (Fig. 1 E). A separate in vitro preparation protocol for Taxol and GMPCPP lattices, assembled from tubulin without any fluorophore or biotin labels, similarly yielded expanded microtubules (Fig. S3, D–F) (Estevez-Gallego et al., 2020; Vale et al., 1994).

We then assessed the lattice spacing of microtubules from untreated U2OS cells using this in situ layer line analysis. In line with in vitro results of GDP-bound microtubules, 74% (23 out of 31) had a compacted average monomer spacing of 40.8–41.1 Å (Fig. 1 E). The remaining microtubules had an expanded spacing of 41.5–42.7 Å. Our results indicate that microtubule lattice spacings are variable even far away from microtubule tips, where an altered lattice spacing is expected (Hyman et al., 1995; LaFrance et al., 2022). Within the field of view of our data, which is ~0.8 μm, we did not observe any microtubule ends. In contrast to the changes in lattice spacing at microtubule ends, which are important for the regulation of microtubule growth dynamics (Manka and Moores, 2018b), the different lattice spacings that we observe in the shaft of the microtubule likely play a role in microtubule stability, MAP binding, and motor protein kinetics, as has been previously postulated (Cross, 2019; Zhang et al., 2018).

Having established a baseline distribution of lattice spacings, we then set out to test if the lattice spacing within cells could be altered by treatment with Taxol to help explain its mechanism of action. Microtubules that are polymerized in vitro in the presence of Taxol are structurally altered compared with the GDP-bound, compacted state (Alushin et al., 2014; Kellogg et al., 2017; Rai et al., 2020). We therefore expected that Taxol would expand microtubules inside cells as well. Indeed, in U2OS cells treated with 1 μM Taxol for 16 h, we observed microtubules with typical morphology (Fig. 2 A) but found that most microtubules (97%, 29/30) had an expanded lattice spacing of 42.3–43.2 Å (Fig. 2 B), significantly different from the untreated distribution (P value <0.0001). Subtomogram averaging showed an average with 13 protofilaments (PFs) and did not reveal any unusual features at our limited 25 Å resolution (Fig. 2 C and Fig. S4 C). The helical rise was approximated at 9.82 Å, corresponding to a monomer spacing of 42.5 Å, which matches the spacing found with the layer line analysis well. Together with the untreated cell data, these data all show that changes in lattice spacing can be detected for individual microtubules with our in situ layer line analysis and that the overall distribution of microtubule lattice spacings is heavily altered by Taxol.

Remarkably, the Taxol-induced expansion to 41.2–42.0 Å observed in vitro (Alushin et al., 2014; Estevez-Gallego et al., 2020; Kellogg et al., 2017; Rai et al., 2020; Vale et al., 1994) is considerably smaller than the average hyper-expansion of 42.6 Å measured using our in situ approach. Our in vitro measurement of ~42.2 Å for the Taxol lattice is also less than what we most commonly observe in cells (P value <0.01). A complex interplay between Taxol, MAPs, and PTMs might exist as up or downregulation of MAPs, like tau, can affect the sensitivity of cells to Taxol (Orr et al., 2003). Additionally, Taxol treatment

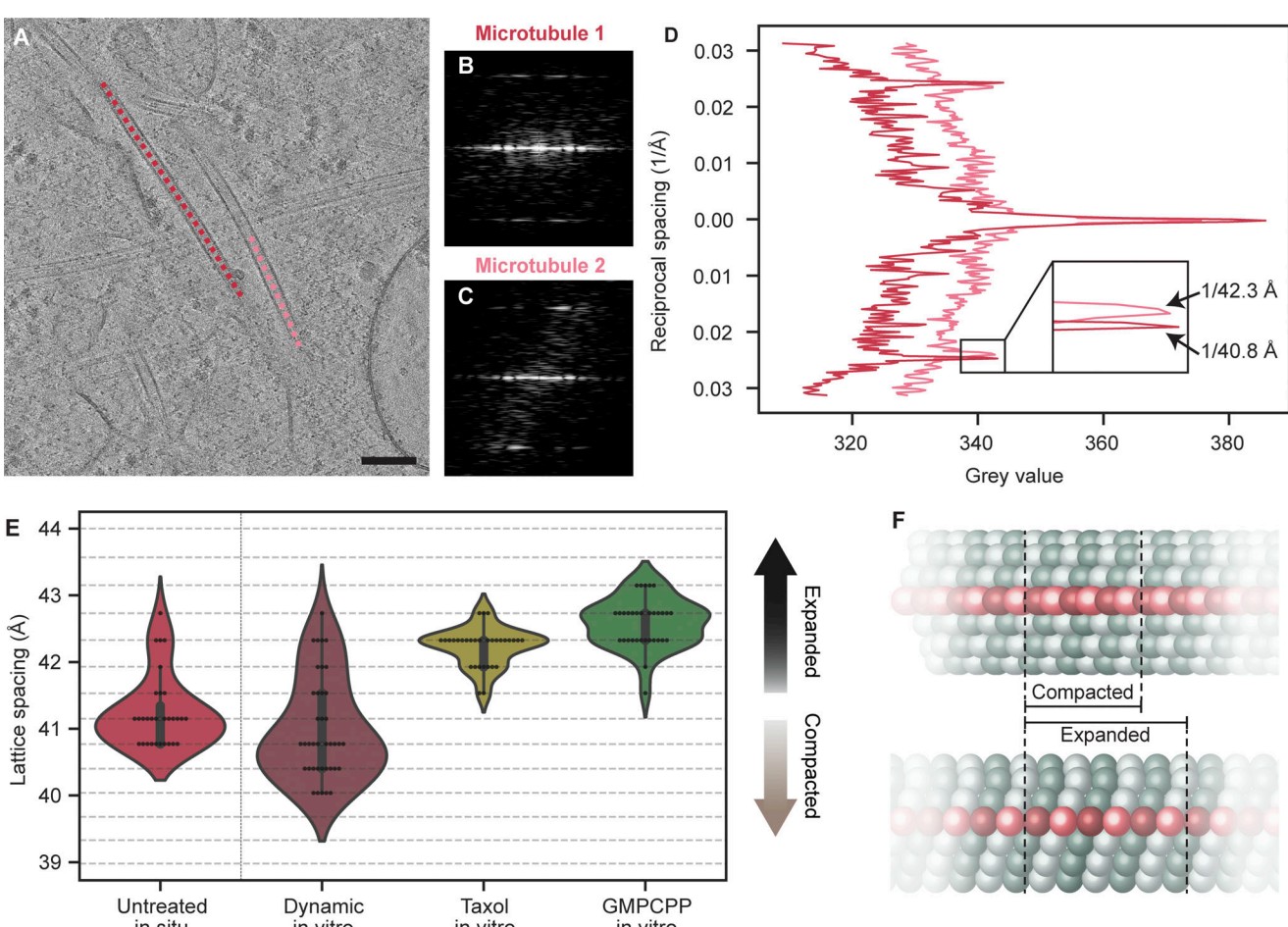

Figure 1. **A subset of microtubules has an expanded lattice in cells. (A)** Tomogram slice (thickness: 10 nm) showing two selected microtubule (MT) backbones in an untreated U2OS cell (red and pink). **(B and C)** Power spectra of the masked and transformed MT segments from the red MT (B) and pink MT (C) shown in A. **(D)** Overlay of the layer line plots of the power spectra of the MT segments from the compacted (red) and expanded (pink) MTs in A. Arrows indicate the location of the layer line peaks and their related lattice spacing. **(E)** Violin plot showing the distribution of lattice spacings in untreated U2OS cells (N = 31, 12 tomograms, 7 cells), from microtubules assembled in vitro from GTP-bound soluble tubulin yielding dynamic microtubules (N = 40, 6 tomograms), in the presence of Taxol (N = 32, 3 tomograms), or from GMPCPP-bound soluble tubulin (N = 33, 14 tomograms). Horizontal lines correspond to the discrete spatial frequency values in reciprocal space. **(F)** Simplified cartoon showing the long-range effect of a compacted or an expanded MT lattice. Scale bar: 100 nm (A).

leads to an increase in PTMs like acetylation (Hammond et al., 2010). These components are usually completely or partially absent from in vitro systems and may amplify each other through positive reinforcement feedback loops within the cell, which may explain why this hyper-expansion has thus far remained unreported.

Having established that different microtubule lattice spacings can be monitored within the cell and that stabilization with Taxol leads to lattice expansion, we next set out to determine if the lattice spacing for the more stable subset of microtubules differed from the well-known compacted state at physiological conditions. For this, we used a kinesin-1 rigor construct, StableMARK (Jansen et al., 2023), which preferentially binds to a subset of stable microtubules that are often highly acetylated and less dynamic.

We set up a two-step cryo-correlative light and electron microscopy (cryo-CLEM) workflow, wherein cryo-fluorescence microscopy (cryo-FM) data of U2OS Flp-In T-Rex cells expressing StableMARK, fluorescently labeled with NeonGreen, was used for both the lamella preparation and ultimate cryo-ET data correlation. Cryo-FM data of cells grown on grids was collected using two different cryo-FM microscopes, either a cryo-FM (Meteor) integrated into the cryo-focused ion beam and scanning electron microscope (cryo-FIB-SEM) or a dedicated, separate confocal cryo-FM (CorrSight), as described in Materials and methods, yielding two slightly different workflows. The integrated cryo-FM allowed us to image the cells after milling lamellae to a thickness of ~400 nm. Only the lamellae that still contained StableMARK at this point were thinned further to a thickness suitable for cryo-ET imaging (~150 nm) (Fig. 3, A–D and I). Fluorescent imaging at this stage allowed us to exclude cells that had lost their StableMARK signal during milling. This increased the workflow's throughput and success rate considerably compared with the CorrSight workflow which only used FM data prior to milling. Given the fixed spatial relationship between the integrated FM and SEM optics, as they are stable

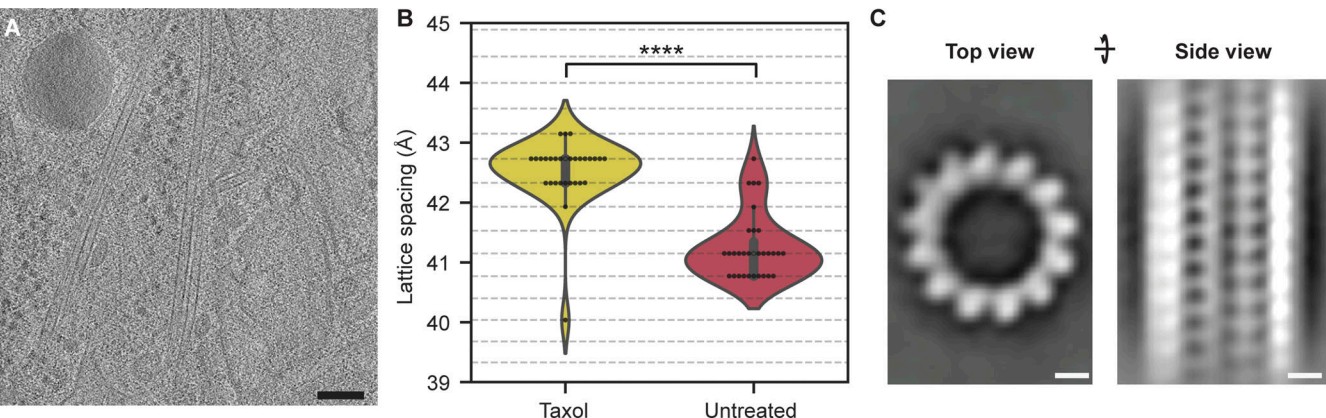

Figure 2. **Taxol treatment induces a hyperexpanded lattice within cells. (A)** Tomogram slice showing a representative image of Taxol-treated micro-tubules in WT U2OS cells. Scale bar: 100 nm. **(B)** Violin plot showing the lattice spacing distribution in Taxol treated cells (*N* = 30, 6 tomograms, 5 cells) and in untreated cells (*N* = 31, 12 tomograms, 7 cells, same data as Fig. 1 E, included for comparison). Horizontal lines correspond to the discrete spatial frequency values in reciprocal space. Taxol distribution is significantly different from the untreated distribution (****P value <0.0001, unpaired *t* test based permutation test). **(C)** Microtubule average shows that Taxol-treated microtubules consist of 13 PFs. Central volume slices (28 nm thick) from top (left) and side (right) views. Scale bar: 5 nm.

parts of the same instrument, FM-SEM correlation could simply be obtained by calculating a scaling factor (Fig. 3 F). This scaling factor was then used to overlay the FM and SEM images (Fig. 3, G–I). The extracellular beads (Fig. 3 G), grid holes, and the edges of the lamella were used to confirm the correlation. FM data of endocytosed fBSA-Au[5] beads (Fermie et al., 2022), present inside the cells, was obtained as well (Fig. 3 H) to be later used to overlay the FM and transmission electron microscopy (TEM) data.

In contrast, when using the separate confocal instrument, cryo-FM data Z-stacks were recorded on intact cells and used to target lamella preparation sites before any thinning was done (Fig. S5 A). This required a more elaborate targeting procedure. The FM stacks were correlated to the SEM images by calculating a 3D transformation matrix using the extracellular beads (Fig. S5, B–F). The FM-SEM localization accuracy for both workflows was estimated using a leave-one-out metric (see Materials and methods) and showed an FM-SEM localization error with an SD of 180 nm in x and 200 nm in y for the Meteor workflow (Fig. 3 E) and 121 nm in x and of 116 nm in y for the CorrSight workflow (Fig. S5 G). This is lower than previously reported SEM localization errors (SD of 225 nm in x and 341 nm in y), which were based on a workflow similar to our CorrSight workflow (Arnold et al., 2016). This is most likely due to the increased number of beads we used for calculating the transform (seven to nine instead of six). The estimated localization errors confirmed that the FM-SEM correlation was sufficiently accurate for the first step of our CLEM approach using either instrument.

In the second step, similar for both workflows, the cryo-FM data was then correlated with the cryo-TEM data to identify StableMARK-bound microtubules (Fig. 4 A). For this step, the fluorescent signal from the endocytosed fBSA-Au[5] beads (Fermie et al., 2022) was correlated with regions in the TEM data where clusters of electron-dense beads were observed (Fig. 4, B–E and

Fig. S6, A–D). The resulting overlay was used to accurately align the StableMARK signal to the cryo-TEM data. Fine overlay adjustments based on the location of the gold beads in the TEM data were performed. This was critical due to the limited resolution of the FM data and because the microtubules alone did not allow for assessment or optimization of the correlation (Fig. 4 F and Fig. S6 E). When the correlation was considered unambiguous and reliable, the StableMARK data was used to localize the StableMARK-bound microtubules within the lamella (Fig. 4, G–I and Fig. S6, F–H). Analysis of their lattice spacings revealed a clear shift toward a more expanded lattice distribution centered around 41.5–41.9 Å (P value <0.0001, N = 43, 12 tomograms), in comparison to the dominant compacted lattice observed for untreated microtubules (Fig. 4 J). Both workflows, using the two different cryo-FM instruments, yielded similar distributions for StableMARK-bound microtubules (Fig. S6 I, P value = 0.091). For undecorated microtubules near the positively identified StableMARK-bound microtubules, we observed compacted lattices (for example, Fig. S7, A–E) and obtained a distribution significantly different from the StableMARK distribution (P value <0.01, N = 12, 10 tomograms, Fig. S7 F), although we cannot exclude very low decoration of these microtubules due to the limited sensitivity of both cryo-FM instruments. The StableMARK distribution was also significantly different from the lattice distribution observed for Taxol-treated cells (P value <0.0001). Our results indicate that the stable, long-lived subset of microtubules inside cells recognized by StableMARK is predominately more expanded than the more dynamic subset.

Despite the additional correlation accuracy gained with the intracellular beads and the integrated cryo-FM, we cannot exclude that a proportion of the StableMARK-bound microtubules were identified incorrectly due to the limited resolution of our cryo-FM setup. This fractional misassignment might contribute to the large spread observed in lattice spacings for the StableMARK-bound microtubule subset. Technical FM improvements such as cryo-MINFLUX (Gwosch et al., 2020) or

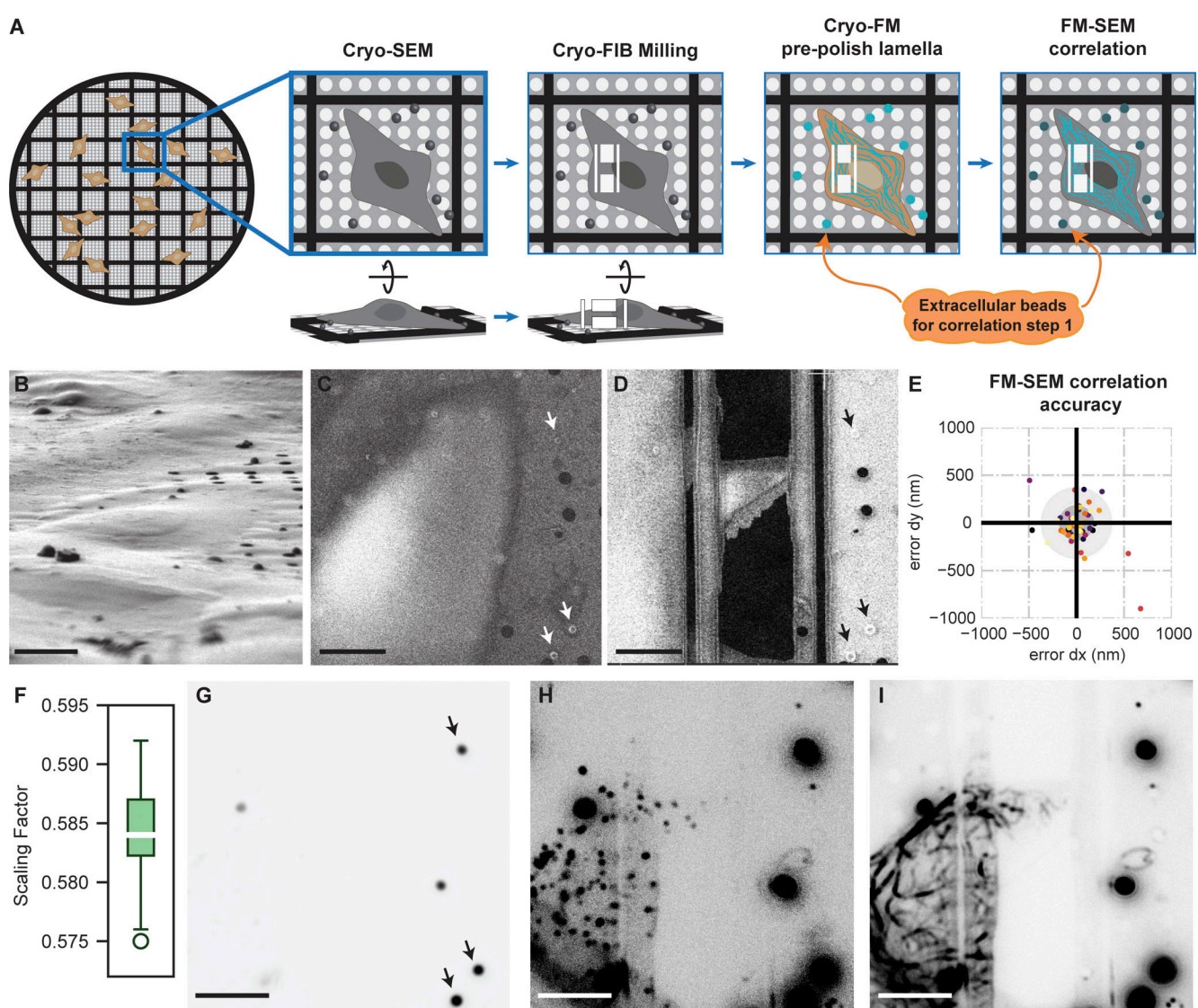

Figure 3.   **Correlation of FM to SEM data using an integrated cryo-FM. (A)** Cartoon describing the FM-SEM correlation with FM data obtained after milling. Correlation is confirmed using extracellular beads. **(B)** FIB image of an intact U2OS cell (9° tilted side view). **(C)** Untilted SEM image of the same grid square as shown in B. The beads used to confirm FM-SEM correlation are indicated with white and black arrows in C and D, respectively. **(D)** Untilted SEM image of the polished lamella of the cell shown in C. **(E)** Scatterplot of correlation errors from leave-one-out calculations; each dataset has a unique color, grey circles mark the 1xSD and 2xSD boundaries (10 datasets, 51 beads), dx = difference in x, dy = difference in y. **(F)** Boxplot showing the distribution of scaling factors (mean = 0.584, standard deviation = 0.005, N = 10). **(G)** Scaled FM image of the extracellular beads used to guide FM-SEM overlay, beads used to confirm FM-SEM correlation are indicated with black arrows, similar to C and D. **(H)** Scaled FM image of fBSA-Au[5] beads used for subsequent FM-TEM correlation (see Fig. 4). **(I)** Scaled FM image of the StableMARK signal. Scalebars: 10 μm (B–D and G–I).

engineered point spread functions (Zhou et al., 2019) may improve the cryo-FM resolution (lateral and/or axial) and resulting CLEM localization accuracy, as well as the throughput of future cryo-CLEM workflows. Nevertheless, the current workflow, which uses an additional independent marker (the fBSA-Au[5] beads) for the FM-TEM correlation, can readily be used to gain new insights into well-studied biological processes from intact cells.

In summary, we present results on microtubule lattice spacing within unperturbed cells, upon Taxol treatment, and in correlation with StableMARK binding. We found that most microtubules have a compacted lattice of ~41 Å, in line with previous in vitro and in situ studies (Fig. S1), but that the stable

microtubules marked by the kinesin rigor construct StableMARK display an expanded lattice more similar to lattice spacings reported in in vitro studies for microtubules assembled using the GTP analog GMPCPP or in the presence of the drug Taxol (Fig. S1). While the interplay between MAPs and microtubule lattice spacing, just as the link between detyrosination activity and microtubule lattice spacing, has recently been explored in several in vitro experiments (Peet et al., 2018; Shen and Ori-McKenney, 2023, *Preprint*; Shima et al., 2018; Siahaan et al., 2022; Yue et al., 2023) (Fig. S1), the relevance of these observations for cellular microtubules had remained unclear given the lack of evidence for co-existing lattice spacings inside cells. Our finding that stable cellular microtubules display

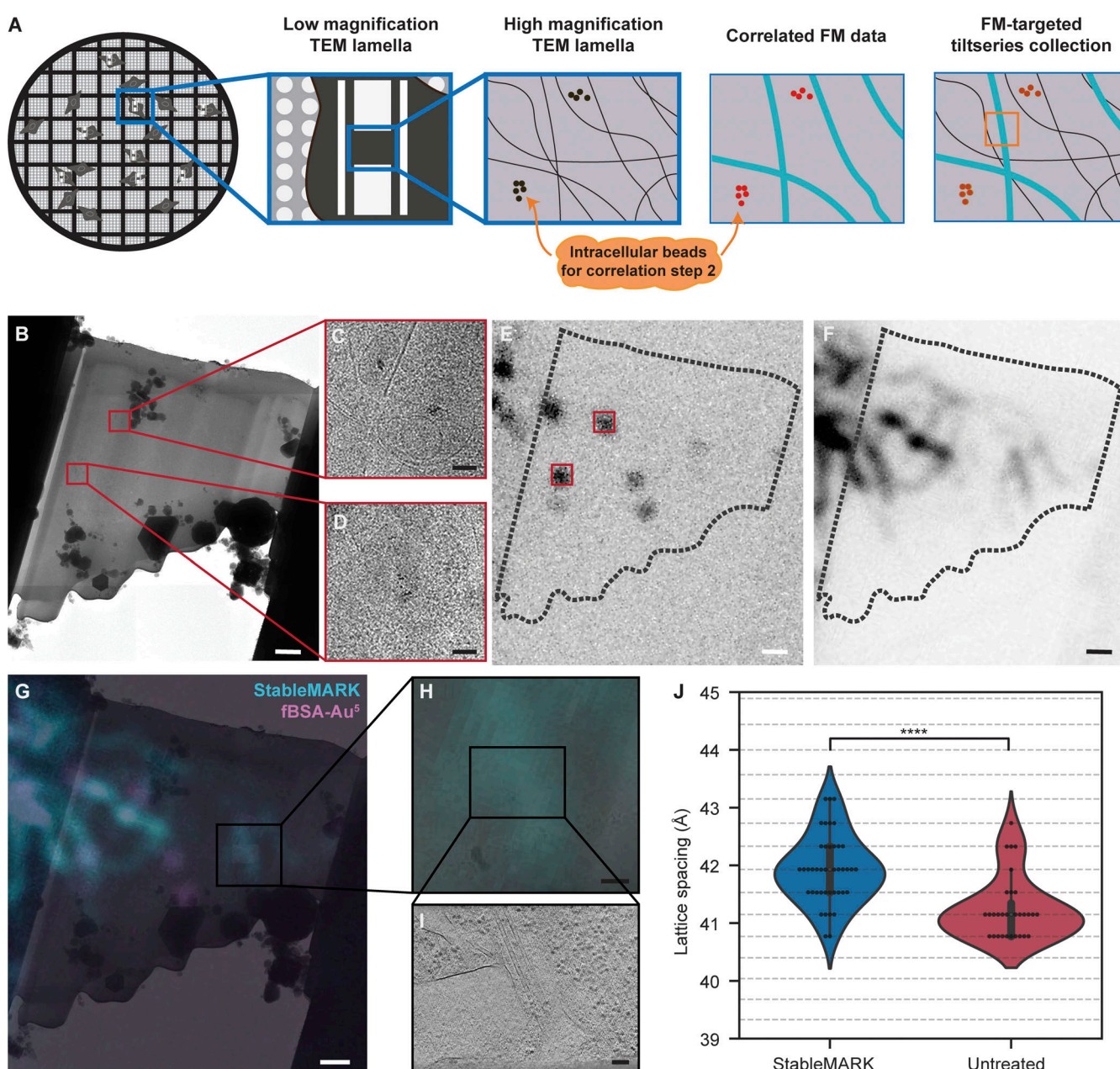

Figure 4. **The lattice of StableMARK-positive microtubules is expanded compared to the GDP-compacted lattice. (A)** Cartoon describing the FM-TEM correlation performed using intracellular beads. **(B)** TEM overview image of a lamella; dotted line indicates outline of the lamella, red squares the location of the fBSA-Au[5] beads. **(C–E)** Zoom-in of the fBSA-Au[5] beads found both in the TEM lamella in B and in the correlated fBSA-Au[5] FM data in E. **(E)** Correlated fBSA-Au[5] FM data; red squares indicate fBSA-Au[5] location. **(F)** Correlated StableMARK-bound microtubules. **(G)** Overlay of TEM lamella and correlated FM data of both fBSA-Au[5] (pink) and StableMARK-bound microtubules (blue). **(H and I)** Two-step zoom of two microtubules overlapping with the StableMARK-bound FM data. **(J)** Violin plot showing the distribution of lattice spacings of the StableMARK subset of microtubules (N = 43, 25 tomograms, 23 cells) and in untreated cells (N = 31, 12 tomograms, 7 cells, same data as Fig. 1 E, included for comparison). Horizontal lines correspond to the discrete spatial frequency values in reciprocal space. StableMARK distribution is significantly different from the untreated distribution (****P value <0.0001, unpaired t test based permutation test). Scale bars: 1 µm (B and E–G), 100 nm (C, D, and I), 400 nm (H).

expanded lattices now directly supports the emerging model that the microtubule lattice geometry functions as an additional cellular mechanism to guide protein binding specificity, on top of the previously established elements of the tubulin code, encouraging further research into the regulatory role of lattice spacing within cells (Akhmanova and Kapitein, 2022; Cross, 2019; Iwanski and Kapitein, 2023; Zhang et al., 2018).

## Materials and methods
Reagents are listed in Table 1.

### Cell lines and cell culture
U2OS wild-type (WT) cells were purchased from ATCC and U2OS Flp-In T-Rex cells were a kind gift from Prof. Alessandro Sartori (Institute of Molecular Cancer Research, University of

Table 1.  **List of reagents**

| Reagent or resource | Source | Identifier |
| --- | --- | --- |
| Q5 HF DNA polymerase | NEB | Cat# M0491L |
| Gibson assembly master mix | NEB | Cat# E2611L |
| DMEM + GlutaMAX-l | GIBCO | Cat# 61965-026 |
| Dulbecco's Phosphate-Buffered Saline (DPBS) | Corning | Cat# 21-031-CV |
| Pen strep | GIBCO | Cat# 15140-122 |
| Trypsin-EDTA | Gibco | Cat# 25200-056 |
| Fetal bovine serum | Sigma-Aldrich | Cat# F7525-500ML |
| Fibronectin | Sigma-Aldrich | Cat# F1141-2MG |
| Blasticidin | Thermo Fisher Scientific | Cat# R21001 |
| Hygromycin B | Corning | Cat# 30-240CR |
| Paclitaxel/Taxol | Thermo Fisher Scientific | Cat# P3456 |
| Paclitaxel/Taxol (for in vitro microtubule reconstitutions) | Enzo Life Sciences | Cat# BML-T104 |
| Doxycycline | Sigma-Aldrich | Cat# 24390-14-5 |
| fBSA-Au[5] | Cell Microscopy Core, Utrecht University Medical Center (UMC) | https://cellmicroscopy.nl/products |
| ProtA-Au[5] | Cell Microscopy Core, Utrecht UMC | https://cellmicroscopy.nl/products |
| Dynabeads MyOne carboxylic acid | Thermo Fisher Scientific | Cat# 65011 |
| Cellview cell culture dish | Greiner bio-one | Cat# 627860 |
| Quantifoil 200 mesh holey carbon R2/2 gold grids | Quantifoil Micro Tools | Not applicable |
| Quantifoil 200 mesh holey carbon R2/2 copper grids | Quantifoil Micro Tools | Not applicable |
| Whatman filter paper | Whatman | Cat# 10311610 |
| Porcine brain tubulin protein (>99% pure) | Cytoskeleton | Cat# T240 |
| Fluorescent HiLyte 647 porcine brain tubulin protein | Cytoskeleton | Cat# TL670M |
| Biotin porcine brain tubulin protein | Cytoskeleton | Cat# 333P |
| X-rhodamine porcine brain tubulin protein | Cytoskeleten | Cat# TL620M |
| PIPES | Sigma-Aldrich | Cat# P6757 |
| MgCl$_2$ | Sigma-Aldrich | Cat# M2670 |
| EGTA | Sigma-Aldrich | Cat# E4378 |
| GpCpp (GMPCPP) | Jena Biocsciences | Cat# NU-405L |
| GTP | Sigma-Aldrich | Cat# G8877 |
| DTT | Thermo Fisher Scientific | Cat# R0861 |

Zurich, Zürich, Switzerland). Cells were confirmed to be free of mycoplasma. The U2OS Flp-In-T-Rex cell line that upon doxycycline-induction expresses hKif5b(1–560)G234A-mNeongreen-

mNeongreen, a kinesin-1 rigor construct, here referred to as StableMARK (Jansen et al., 2023), was derived from the U2OS Flp-In-T-Rex cell line by transfection with the pCDN5/FRT/TO vector (Invitrogen) and pOG44 vector (Invitrogen). The U2OS StableMARK cell line was cultured in Dulbecco's Modified Eagle's Medium + GlutaMAX-l (DMEM-Glu) supplemented with 10% FBS, 100 U/ml penicillin, 100 μg/ml streptomycin, 15 μg/ml blasticidin S, and 0.25 mg/ml hygromycin B. To induce low-level expression of StableMARK, doxycycline (10 ng/ml) was added to the cells 24 h before plunging. U2OS WT cells were cultured in DMEM-Glu supplemented with 10% FBS, 100 U/ml penicillin, and 100 μg/ml streptomycin. Cells were kept at 37°C and 5% $CO_2$.

## Plasmids and cloning

To generate a stable, isogenic U2OS Flp-In-T-Rex cell line, StableMARK was subcloned into pCDNA5/FRT/TO (Invitrogen) via Gibson Assembly using the primer set 5′-GCTCGGATCCACTAG TCCAGTGTGGTGGAATTCTGCAGATGCCACCATGGCGGACCT-3′ and 5′-ACGGGCCCTCTAGACTCGAGCGGCCGCCACTGTGCT GGATGCGGCCGCTTACTTGTACAG-3′. The G234A rigor mutation was used as initially described (Rice et al., 1999). mNeongreen (Shaner et al., 2013) was provided by Allele Biotechnology. The FLP recombinase expression vector is encoded in pOG44 (Invitrogen).

## In vitro microtubule sample preparation

GMPCPP-stabilized microtubules were prepared by incubating 20 μM unlabeled porcine brain tubulin (dataset 8) or a tubulin mix containing 12% HiLyte Fluor 647–labeled tubulin and 18% biotinylated tubulin (dataset 5), with 1 mM GMPCPP in MRB80 (80 mM K-PIPES, pH 6.8, 4 mM MgCl$_2$, 1 mM EGTA) for 30 min at 37°C. Polymerized microtubules were pelleted through centrifugation at 199,000 × $g$ for 5 min, using an Airfuge (Beckman Coulter). Next, the microtubules were gently resuspended in new MRB80 and depolymerized on ice for 20 min. Subsequently, microtubules were repolymerized again with newly added 1 mM GMPCPP. Polymerized microtubules were pelleted as described above and diluted in 50 μl MRB80 buffer containing 1% glycerol. Finally, the GMPCPP-stabilized microtubules were flash-frozen and stored at –80°C, except for the unlabeled sample (dataset 8), which was plunged directly after preparation.

Taxol-stabilized microtubules were prepared by incubating 29 μM unlabeled porcine brain tubulin (dataset 7) or a mix of 25 μM unlabeled porcine brain tubulin, 1.25 μM X-rhodamine porcine tubulin, and 2.8 μM biotinylated porcine tubulin (dataset 4) in the presence of 2.5 mM GTP and 20 μM Taxol for 30 min at 37°C. Next, an excess of 20 μM Taxol was added and the polymerized microtubules were pelleted at room temperature through centrifugation at 16,200 × $g$ for 15 min using a table-top centrifuge. The pelleted Taxol-stabilized microtubules were gently resuspended in 50 μl MRB80 buffer containing 20 μM Taxol, stored at room temperature, and used for cryo-EM sample preparation within the same day.

Dynamic microtubules (dataset 6) were grown through spontaneous nucleation by incubating 45 μM porcine brain

tubulin in MRB80, supplemented with 1 mM GTP and DTT, in an Eppendorf tube for 15 min at 30°C. ProtA-Au[5] fiducials (1:30) were added to the dynamic and stabilized microtubule samples and stabilized microtubules were diluted 16-fold in warm MRB80, prior to plunge-freezing. The samples were gently resuspended and 4 µl was pipetted onto glow-discharged holey carbon R2/2 copper grids (Quantifoil Micro Tools). The grids were manually back-blotted for 3 s and the samples were subsequently vitrified in liquid ethane, after which they were clipped into cartridges and stored in liquid nitrogen until further use.

**In situ sample preparation**
To remove toxic manufacturing residues, Quantifoil 200 mesh holey carbon R2/2 gold grids (Quantifoil Micro Tools) were shortly washed in chloroform, ethanol, methanol, and Milli-Q water (increasing polarity with every step) after which they were UV-sterilized. On the day of cell seeding, the grids were glow-discharged (PELCO easiGlow; Ted Pella) and placed on 40 µl fibronectin droplets (50 µg/µl) and incubated at 37°C for 2–3 h. Subsequently, the grids were washed two times by placing them on droplets of 40 µl PBS and put in 35-mm glass bottom dishes (Greiner Bio-One). 90,000 cells in 2 ml medium were seeded on these grids and were left to settle in the hood for ~20 min before placing them in the incubator. After 24 h, the U2OS StableMARK cells were treated with doxycycline (10 ng/ml). WT U2OS cells were treated with Taxol (1 µM) 16 h prior to plunging, if applicable. Plunging was performed 48 h after cell seeding. In preparation for vitrification, the cells were incubated with fBSA-Au[5] (Cell Microscopy Core Utrecht UMC, diluted in DMEM to OD 5) for 3–4 h (Fermie et al., 2022). Just before plunging, the media was exchanged for fresh DMEM. The grids were washed by dipping two times in PBS (37°C) before 3 µl of 1 µm Dynabeads (Thermo Fisher Scientific: MyOne with 40% iron oxide, carboxylic acid) diluted 1:20 in PBS was added to the grids. Finally, the cells were vitrified in liquid ethane after manually blotting for 10 s. The grids were clipped into cartridges and kept at liquid nitrogen temperature throughout the subsequent experiments.

**SEM grid screening (for CorrSight-based workflow only)**
To increase the efficiency of the CorrSight-based cryo-CLEM workflow, although at the cost of ice contamination, grids were screened in the cryo-FIB-SEM (Aquilos; Thermo Fisher Scientific) prior to fluorescent imaging, SEM grid overview images were taken using FEI MAPS 3.8 software. Grids with clearly visible grid holes, indicative of appropriate blotting, and appropriate distribution of cells were used in the next steps of the cryo-CLEM workflow.

**Cryo-FM (for CorrSight-based workflow only)**
Cryo-FM data were obtained prior to milling with the FEI CorrSight equipped with a cryo-stage using the Andromeda spinning-disk confocal microscope module with Hamamatsu ORCA-Flash4.0 camera. Grid overviews were collected with an EC "Plan-Neofluar" 5×/0.16 NA air objective using transmitted light. Previously obtained SEM overview images were aligned to

the FM overview images via three-point correlation in the MAPs v3.8 software (Thermo Fisher Scientific). Based on the SEM and FM overlay, candidate cells were selected. Using the EC Plan-Neofluar 40×/0.9 NA air objective, z-stacks ranging from 8 to 14 µm with 300-nm step size were collected. Z-stacks were recorded to capture three different fluorescent probes, namely StableMARK (488 nm), Dynabeads (488 nm), and fBSA-Au[5] (561 nm). Images were recorded with FEI MAPS v3.8 software and LA FEI Live Acquisition v2.2.0 (Thermo Fisher Scientific). The images were subjected to a mild deconvolution using Huygens Professional software v. 21.04 (Scientific Volume Imaging) with a classic maximum likelihood estimation algorithm.

**Targeted cryo-FIB milling**
FM-guided ion beam milling was either performed using FM data obtained with the FEI CorrSight (discussed in the section above) or FM data obtained during the milling session using the Meteor system (Delmic). Cryo-FIB milling was performed in the cryo-FIB-SEM (Aquilos; Thermo Fisher Scientific). Lamellae were prepared as established by Wagner et al. (2020). In preparation for milling, the grids were coated with platinum to reduce charging effects. Eucentric heights and minimal stage tilt angles (16–18°, corresponding to 9–11° lamella angle), to ensure access of the ions to the milling sites, were determined.

Next, in cases where CorrSight-based FM data (recorded prior to milling) was used, SEM grid overview images were obtained and overlayed with the cryo-FM overview images using the three-point correlation tool available in the MAPS v3.8 software to guide the localization of candidate lamella sites. High-magnification SEM images (0.135 µm/pixel, 1 µs dwell time, 1,536 × 1,024 pixels, 2 keV, 13 pA) of each target cell were taken and manually overlayed with the corresponding cryo-FM maximum intensity projection (MIP) using Dynabeads as fiducials. The location of the milling patterns was based on the correlated cryo-FM data. An FM-FIB correlation was not included in our workflow as this is a time-intensive procedure and the limited z-resolution of our cryo-FM setup meant this correlation did not add significant information when tested.

Subsequently, the grids were subjected to organo-platinum deposition for 10 s via an integrated gas injection system to generate a more even surface and thereby reduce curtaining effects and protect the final lamella. Milling was performed with a stepwise decreasing current of 1 to 0.3 to 0.1 nA and a shrinking milling pattern. Microexpansion joints were added during the first step to increase lamellae stability (Wolff et al., 2019). The final polishing step was performed at 30 pA to reach a final lamella thickness of 80–140 nm. High-magnification SEM images of the polished lamellae were taken (0.135 µm/pixel, 300 ns dwell time, 1,536 × 1,024 pixels, 2 keV, 13 pA) and the grids were coated with platinum for a second time before unloading.

Alternatively, in case the Meteor system was used, milling was performed without prior FM knowledge, and FM data of the lamellae was obtained before polishing. The grids were subjected to organo-platinum deposition for 10 s directly after the eucentric height was determined. Milling was performed with a stepwise decreasing current of 1 to 0.3 to 0.1 nA and a shrinking milling pattern generating lamella of ~400 nm. Next, the

lamellae were imaged with the integrated Meteor system (Delmic). Using an Olympus LMPLFLN 50×/0.5 NA air objective (without chromatic aberration correction), Andro Zyla sCMOS 4.2 camera, and the software Odemis 3.2.1, Z-stacks of 10 μm with a step size of 500 nm were collected. Two different z-stacks were recorded to capture three different fluorescent probes within the lamellae, namely StableMARK (LED excitation of 470 nm, emission filter 515/30), Dynabeads (LED excitation of 470 nm, emission filter 515/30), and fBSA-Au[5] (LED excitation of 555 nm, emission filter 595/31). The final polishing step was performed at 30 pA to reach a final lamella thickness of 100–200 nm. This was only done for the lamellae that still contained the target of interest according to the collected FM data. High-magnification SEM images of the polished lamellae were taken (0.135 μm/pixel, 300 ns dwell time, 1,536 × 1,024 pixels, 2 keV, 13 pA) and the grids were coated with platinum for a second time before unloading.

## Targeted cryo-ET

In preparation for FM-guided cryo-ET data collection, SEM images of polished lamellae were aligned to the corresponding cryo-FM z-stacks, guided by high-dose SEM images of the lamella sites. In case CorrSight-acquired FM data was used, six to nine Dynabeads were localized both in the SEM image and the FM z-stack using the 3D Correlation toolbox (Arnold et al., 2016). Next, a transformation matrix was fitted for the two sets of x, y, and z coordinates, while aiming for a root mean square deviation <1 pixel for each bead. Z-stacks were transformed according to this matrix with the 3D rigid body transformation of the Pyto Python package (Arnold et al., 2016). MIPs of the transformed z-stack of each channel were overlayed with the SEM image of the polished lamella in FIJI (Schindelin et al., 2012) to generate the final correlated FM-SEM overlay.

In cases where Meteor-acquired FM data was used, FM images could simply be scaled with a predetermined scaling factor, as the SEM and FM imaging systems are at a fixed position with respect to each other. To do so, the average scaling factor of 10 lamella sites was calculated using again the 3D correlation toolbox. This yielded an average scaling factor of 0.584 ± 0.005. This predetermined scaling factor was subsequently used for all lamellae and was applied in FIJI. The scaled FM image and SEM image were manually positioned in PowerPoint using the Dynabeads and lamella shape to generate the final correlated FM-SEM overlay.

Lamellae were imaged on a 200 kV Talos Arctica transmission electron microscope (TEM) (Thermo Fisher Scientific) with a K2 summit direct electron detector (Gatan) or on a 300 kV FEI Titan Krios TEM (Thermo Fisher Scientific) with a K3 summit direct electron detector (Gatan), both equipped with a post-column energy filter aligned to the zero-loss peak and a 20 keV slit width. Using SerialEM (Mastronarde, 2003), lamellae overview images were taken at 7,300× (Arctica, 18.7 Å/pixel) or 4,800× (Krios, 38.6 Å/pixel).

The TEM overview image of the lamella and the corresponding FM-SEM overlay were further manually correlated. This last correlation step was guided by the bimodal intracellular fBSA-Au[5] fiducials and the lamella shape. In cases where the

StableMARK FM signal overlapped with a microtubule in the cryo-TEM lamella image, a tilt series was collected. The microtubules in the lamellae of untreated and Taxol-treated cells were chosen at random. Tilt series of microtubules were recorded at a pixel size of 2.17 Å/pixel, a dose rate of ~5 (Arctica) or 10–20 (Krios) e−/pixel/s, and a total dose of 90–100 e−/Å[2]. All tilt series were collected using a dose symmetric scheme (Hagen et al., 2017), a tilt increment of 2° or 3°, a defocus target of 2.3 μm, and generally with a tilt range of 63° to −45° or 45° to −63°, depending on the lamella orientation in the microscope. Tilt series of in vitro microtubules were collected across ranges of 54° to −54° or 60° to −60°.

## Tomogram reconstruction

The tilt series' movies were aligned and dose-weighted using MotionCor2 (Zheng et al., 2017). The tilt series were generally aligned via patch tracking. If fBSA-Au[5] beads were present, fiducial tracking was used. Tomogram reconstruction was performed in Etomo, part of the IMOD 4.10.29 package (Kremer et al., 1996). Contrast transfer function estimation and correction was performed in IMOD using the ctfplotter and ctfphase-flip commands and the tomograms were reconstructed using weighted back-projection and a Simultaneous Iterative Reconstruction Technique (SIRT)-like filter with 3 or 20 iterations, for in situ or in vitro data, respectively.

## Layer line analysis

Reconstructed tomograms were loaded in Dynamo (Castano-Diez et al., 2017). The filament model (crop along axis) was used to pick the microtubule backbone. The backbone coordinates were exported from MATLAB and used to generate a soft mask around the microtubule. Particles with a box size of 190 pixels were extracted from the unbinned, masked tomograms (2.17 Å/pix). After cropping according to the filament model, the particles were inspected using the dGallery graphical user interface. Per microtubule, multiple aligned particles were selected to cover the whole microtubule, with minimal overlap (~10%). These particles were extracted from the unbinned tomograms with a box size of 1,030 pixels (224 Å), minimally 26 heterodimers long, and summed along the volume z-axis. The power spectrum of each particle was calculated in FIJI. The power spectra of the particles from the same microtubule were summed to increase the signal-to-noise ratio. Layer lines were localized by calculating a line profile plot of the power spectrum. The average lattice spacing of the microtubule was calculated using Eq. 1.

$$Lattice\ spacing = \frac{pixel\ size\left(\frac{Å}{pix}\right) \times box\ size}{|Equator\ location\ (pix) - Layer\ line\ location\ (pix)|}$$
(1)

This analysis pipeline is visualized in Fig. S2 A.

## Subtomogram averaging

Subtomogram averaging was performed in Relion-4.0 (Zivanov et al., 2022). Tomograms were reconstructed with Relion, based on the previously obtained tilt alignment parameters in IMOD

(Tomogram reconstruction section). Microtubule coordinates were manually picked in IMOD, using overlapping boxes (subtomograms) spaced approximately every dimer (84 Å). The initial rotational angles, around the microtubule axis, were randomized to reduce the missing wedge effect. 4,200 subtomograms at a downsampling factor of 4 (8.68 Å/pixel) and a box size 64 pixels were extracted in Relion. An initial 3D refinement, using a hollow tube (Fig. S4 A) as a starting reference, yielded an average with 13 PFs. This was confirmed by overlaying and cross-correlating the average with the six references (11–16 PFs) from the Microtubule RELION-based Pipeline (MiRP) (Cook et al., 2020). Attempts to sort subtomograms into different PF classes using two, three, or six classes yielded misaligned averages, identical classes, or all particles going to a single, 13 PF class. Next, a helical parameter search was performed around known 13 PF twist and rise values (twist: –27.7°, rise: 9.4 Å) in a 3D classification run (1 class, twist search range: –26.7 to –28.7° and rise search range: 8.4–10.4 Å) using a 40 Å low-pass filtered 13 PF reference (Fig. S4 B) from MiRP (https://github.com/moores-lab/MiRP/tree/master/). A final 3D refinement was performed where the optimized rise (–27.7°) and twist (9.82 Å) values were used to apply helical symmetry. This yielded an average with a resolution of ~25 Å.

### Statistical analyses

Leave-one-out calculations were performed as described by Schorb and Briggs (2014). Briefly, a transformation matrix with the 3D correlation toolbox was calculated for a set of beads. For the CorrSight transforms, seven to nine beads were used while for the Meteor transforms four to eight beads were used. This difference was due to the smaller field of view of the Meteor compared to the CorrSight (diagonal field of view length of 120 versus 277 μm). The transform was applied to a bead not included in the initial set, yielding a predicted bead position. The deviation of the predicted bead position from the true bead position was used to calculate the standard deviation of the errors, which was used as an accuracy measurement.

P values for comparison of the different lattice spacing distributions were calculated with an unpaired two-tailed $t$ test–based permutation test with 10,000 iterations. Permutation tests were performed in Python using the statistical module of the SciPy library (Virtanen et al., 2020). Graphs (boxplots and violin plots) were created with Jupyter notebook 6.0.3 (Kluyver et al., 2016).

Signal-to-noise ratios were calculated in python using the Numpy package, according to Eq. 2 (Unser et al., 1987), where signal refers to the x-axis value of the layer line plots (grey value).

$$SNR = \frac{mean(signal)^2}{std(signal)^2} \qquad (2)$$

### Online supplemental material

Fig. S1 visually summarizes the different lattice spacings that have been reported in relation to different small molecules and proteins. Fig. S2 shows the analysis pipeline used to determine the lattice spacings and the signal-to-noise ratio of the different

box sizes tried. Fig. S3 shows example images of the in vitro microtubule data together with the lattice spacings measured for unlabeled Taxol and GMPCPP-bound microtubules. Fig. S4 shows the masks used for the averaging workflow and the Fourier shell correlation (FSC) curve of the final average. Fig. S5 shows an example of an FM to SEM correlation obtained using the CorrSight workflow. Fig. S6 shows an example of an FM to TEM correlation obtained using the CorrSight workflow, including a comparison of measured lattice spacing values between the CorrSight and Meteor workflow. Fig. S7 shows an example and the distribution of the lattice spacing of microtubules devoid of StableMARK signal. Table S1 provides the references of the data shown in Fig. S1. Table S2 provides the electron microscopy metadata of the different datasets.

### Data availability

The data underlying all plots and figure panels displayed in this paper have been deposited in DataverseNL at https://doi.org/10.34894/0WP537 and are publicly available.

## Acknowledgments

We thank Alessandro Sartori (Institute of Molecular Cancer Research, Zurich, Switzerland) for providing us with the U2OS Flp-In T-Rex stable cells, Vladan Lucic (Max Planck Institute of Biochemistry, Planegg, Germany) and Mihajlo Vanevic (Utrecht University, Utrecht, Netherlands) for computational support, Mariska Gröllers-Mulderij (Utrecht University) for guidance in cell culture, Rutger Hermsen (Utrecht University, Utrecht, Netherlands) for input on statistical data analysis, Mathieu Baltussen (Radboud University, Nijmegen, Netherlands) for help with figure design, and Nalan Liv (UMC Utrecht, Utrecht, Netherlands) for providing us with the intracellular fBSA-Au[5] beads.

This work was supported by the European Research Council (ERC) Consolidator grant 724425 (to F. Förster) and ERC Consolidator grant 819219 (to L.C. Kapitein) and was part of the research program National Roadmap for Large-Scale Research Infrastructure 2017–2018 with project number 184.034.014, which is (partly) financed by the Dutch Research Council (NWO). K.I. Jansen was funded by NWO (Graduate program project 022.006.001). A. Akhmanova and R. Hoogebeen are funded by NWO (project OCENW.XL21.XL21.048). For the purpose of open access, a CC BY public copyright license is applied to any Author Accepted Manuscript version arising from this submission. Imaging was performed at the Utrecht University EM Centre with support from C. Schneijdenberg, J. Meeldijk, and M. Bergmeijer, and the Netherlands Electron Microscopy Infrastructure (grant 84.034.014) helped support access to the Netherlands Center for Electron Nanoscopy with support from operators Wen Yang and Willem Noteborn.

Author contributions: L. de Jager: Conceptualization, Data curation, Formal analysis, Investigation, Methodology, Project administration, Validation, Visualization, Writing - original draft, Writing - review & editing, K.I. Jansen: Methodology, Resources, Writing - review & editing, R. Hoogebeen: Data curation, Formal analysis, Investigation, Validation, Visualization,

Writing - original draft, Writing - review & editing, A. Akhmanova: Funding acquisition, Resources, Supervision, Writing - review & editing, L.C. Kapitein: Conceptualization, Methodology, Supervision, Writing - review & editing, F. Förster: Conceptualization, Funding acquisition, Supervision, Writing - review & editing, S.C. Howes: Conceptualization, Funding acquisition, Project administration, Supervision, Writing - review & editing.

Disclosures: The authors declare no competing interests exist.

Submitted: 30 June 2022

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

# Supplemental material

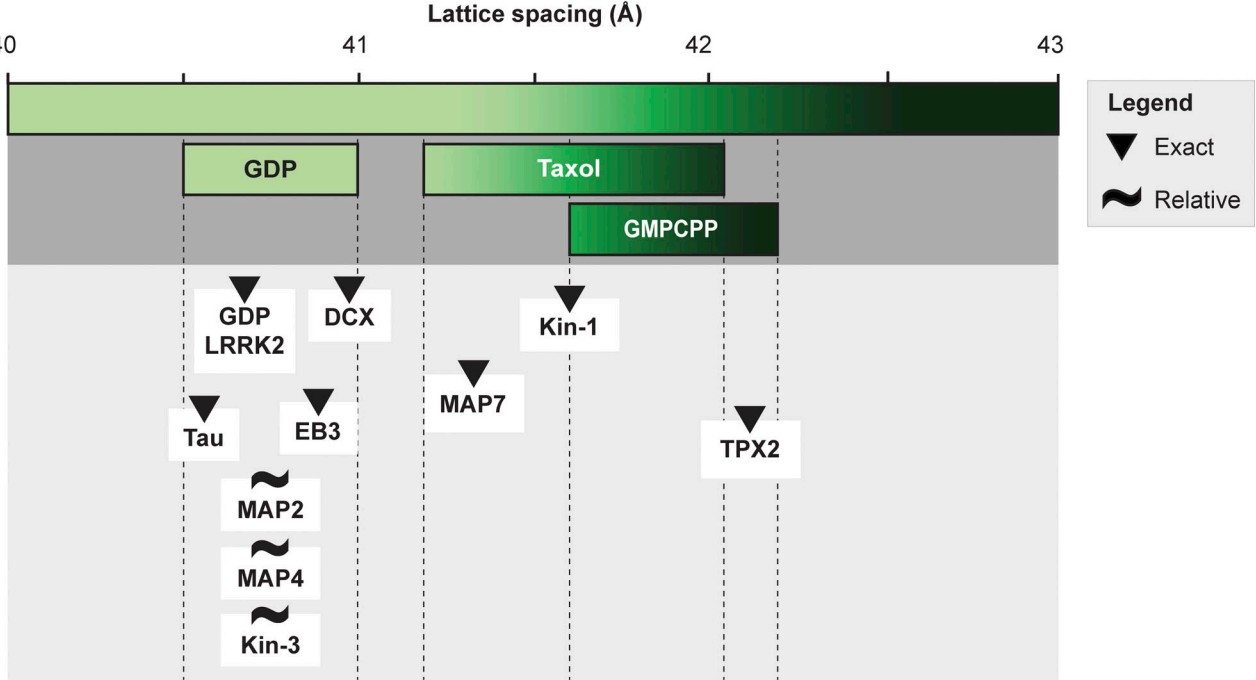

Figure S1. **Summary of previously reported lattice spacings under various conditions.** The figure shows at the top the range of the reported lattice spacings for the GDP, Taxol, and GMPCPP lattices in vitro. Underneath, along the same scale, proteins for which binding in relationship to the microtubule lattice spacing has been investigated are depicted. Their placing is based either on direct (triangle) or relative (tilde) lattice measurements. References in Table S1.

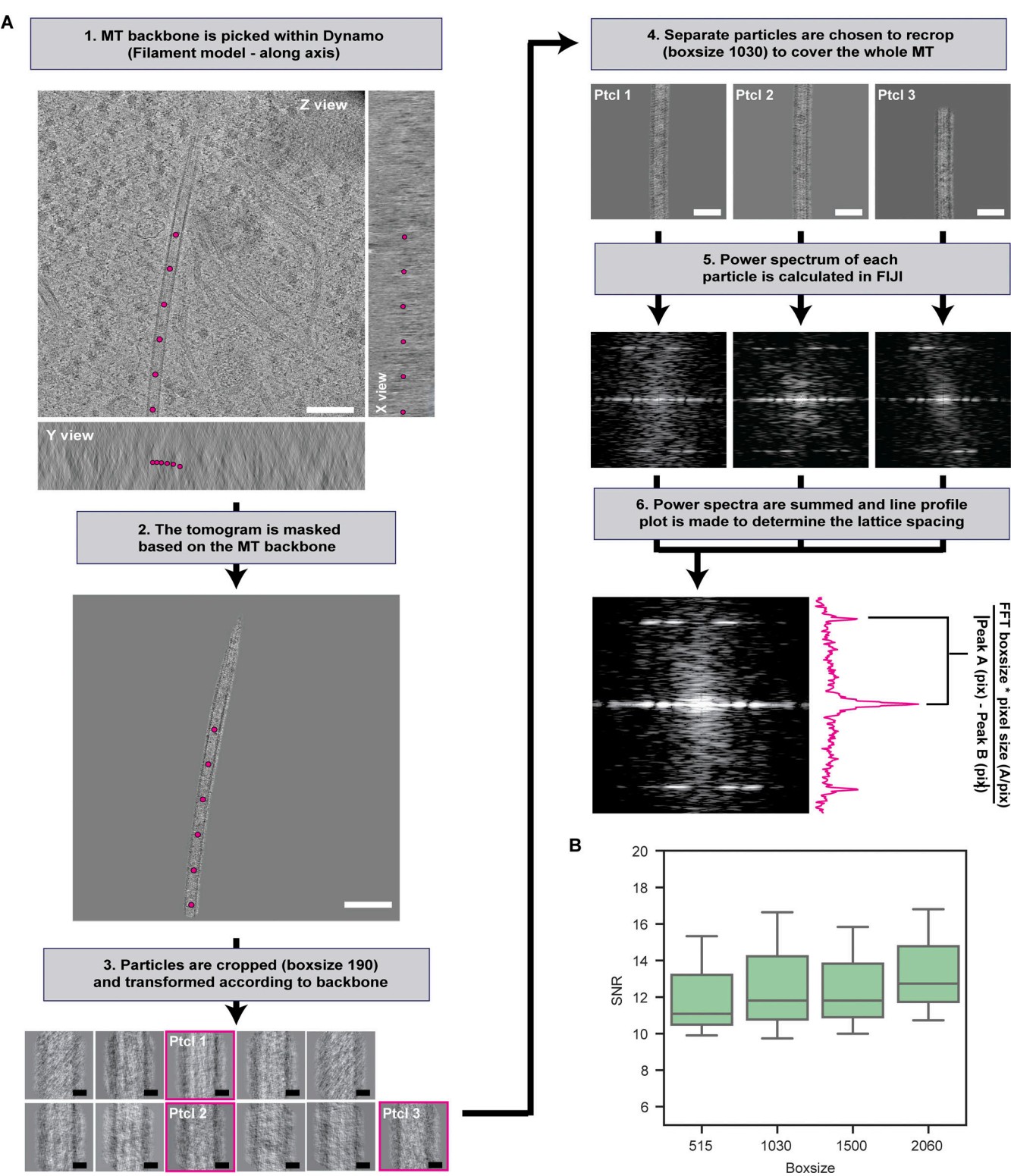

Figure S2. **In situ layer line analysis workflow. (A)** Graphical depiction of the steps performed during the layer line analysis. (Step 1) Microtubule (MT) backbone is traced with evenly spaced coordinates. (Step 2) Using the backbone coordinates, a mask around the microtubule is generated and (Step 3) particles are cropped and transformed to ensure the MT axis is perpendicular for viewing direction. From the cropped particles (box size 190), particles are picked (pink squares) so that after re-cropping, the new particles with a box size of 1,030 (Step 4) cover the whole microtubule with minimal overlap. (Step 5) 2D power spectrum of each particle is calculated. (Step 6) Power spectra of all particles from the same microtubule are summed and the final power spectrum is used to localize the layer lines and thereby calculate the lattice spacing. **(B)** Boxplot of the signal-to-noise ratio (SNR) of the layer line analysis when using different particle box sizes (N = 3). Scale bars: 100 nm (step 1, 2), 10 nm (step 3), 50 nm (step 4).

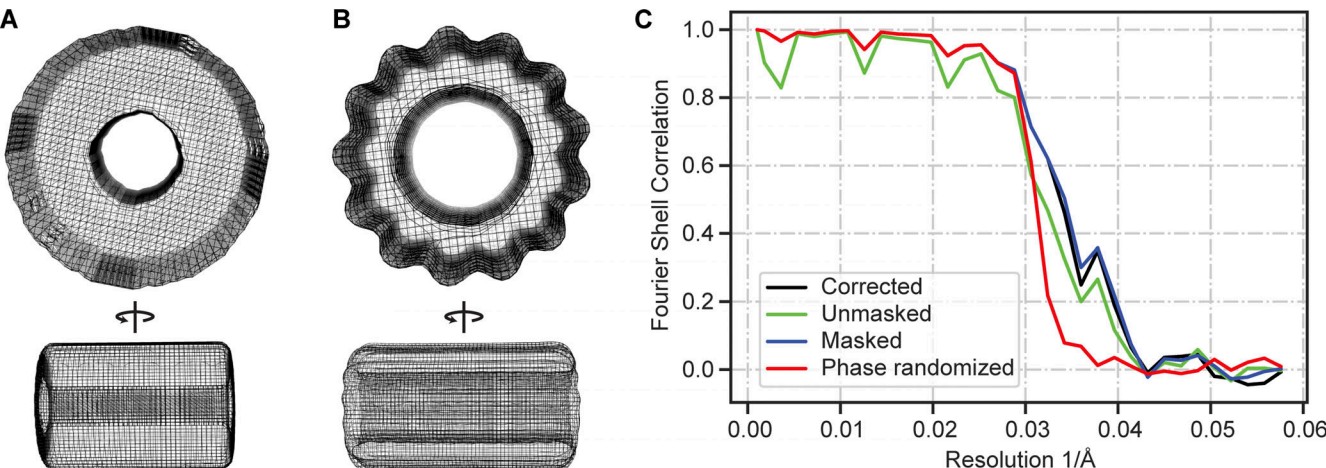

Figure S3. **Representative in vitro microtubule data. (A–E)** Tomogram slices (0.87 nm thick) showing representative images of microtubules assembled in vitro (A) from GTP-bound soluble tubulin yielding dynamic microtubules, (B and C) in the presence of Taxol, either labeled with a fluorophore and biotin (B) or unlabeled (C) (see Materials and methods), or (D and E) from GMPCPP-bound soluble tubulin, either labeled with a fluorophore and biotin (D) or unlabeled (E) (see Materials and methods). Scale bars: 100 nm (A–E). **(F)** Violin plots showing the lattice spacing distribution of unlabeled in vitro Taxol ($N = 10$, 2 tomograms) and unlabeled in vitro GMPCPP-bound microtubules ($N = 10$, 2 tomograms). Horizontal lines correspond to the discrete spatial frequency values in reciprocal space.

Figure S4. **Data analysis components of the in situ Taxol microtubule averaging workflow. (A)** Front view and side view (90° rotation) of the hollow tube reference used for the initial 3D refinement to determine the average PF number. **(B)** Front view and side view (90° rotation) of the 13 PF reference used during subsequent 3D classification and 3D refinement steps. **(C)** FSC curves of the in situ Taxol microtubule subtomogram average are shown in Fig. 2.

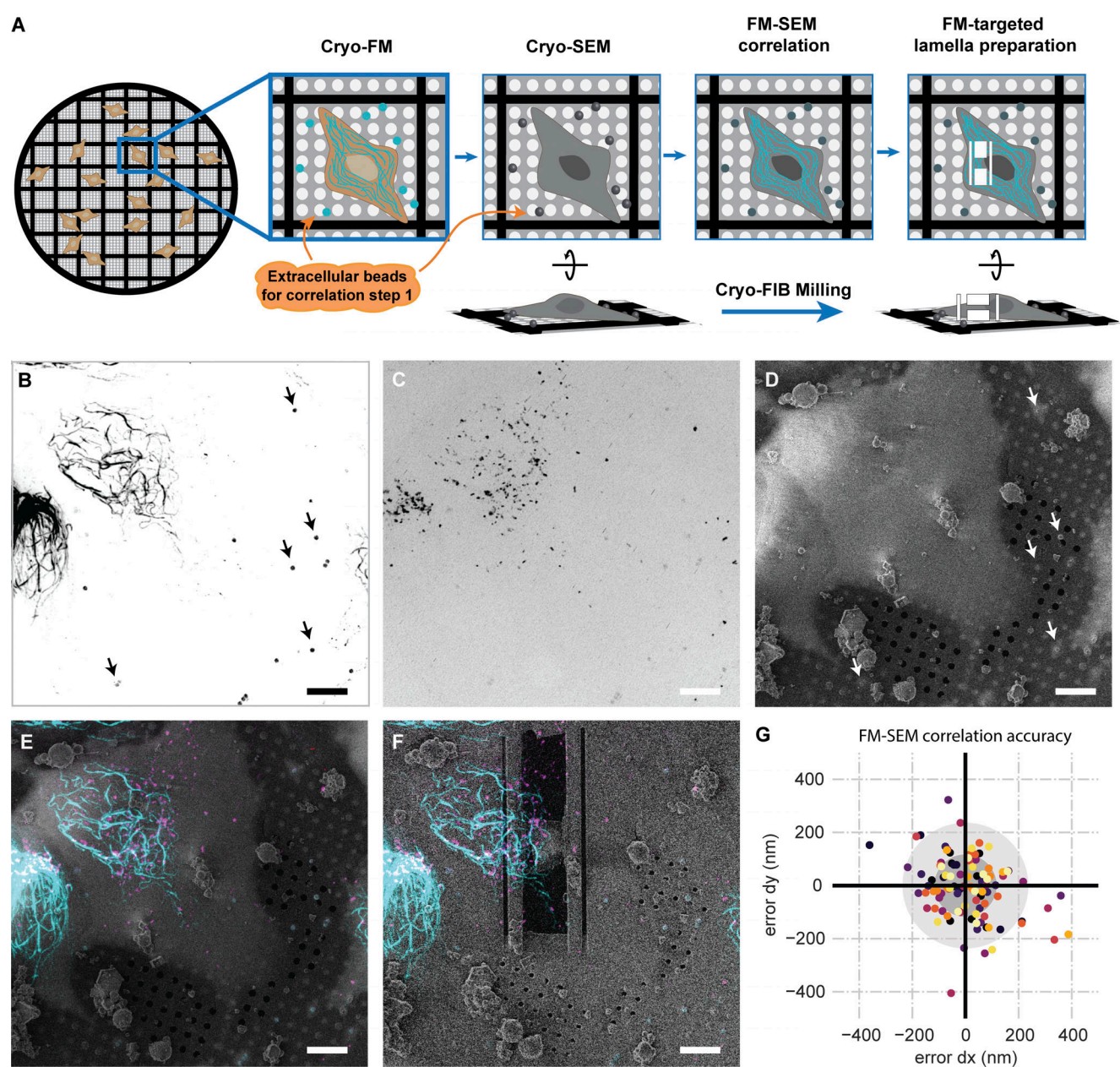

Figure S5. **Correlation of FM to SEM data for targeted cryo-FIB milling. (A)** Cartoon describing the FM-SEM correlation performed using extracellular beads. FM data was obtained prior to milling using the CorrSight. **(B)** MIP of a StableMARK z-stack; beads used for 3D correlation are indicated with arrows. **(C)** MIP of a z-stack with fBSA-Au[5] beads used for the subsequent FM-TEM correlation (see Fig. S6). **(D)** Untilted SEM image of the same grid square as shown in B and C. **(E)** Correlated StableMARK and fBSA-Au[5] overlayed with the SEM image. **(F)** Correlated StableMARK and fBSA-Au[5] overlayed with the untilted SEM image of the polished lamella (milled at a 9° angle). **(G)** Scatterplot of correlation errors from leave-one-out calculations; each dataset has a unique color, grey circles mark the 1xSD and 2xSD boundaries (12 datasets, 98 beads). Scale bars: 10 µm (B–F).

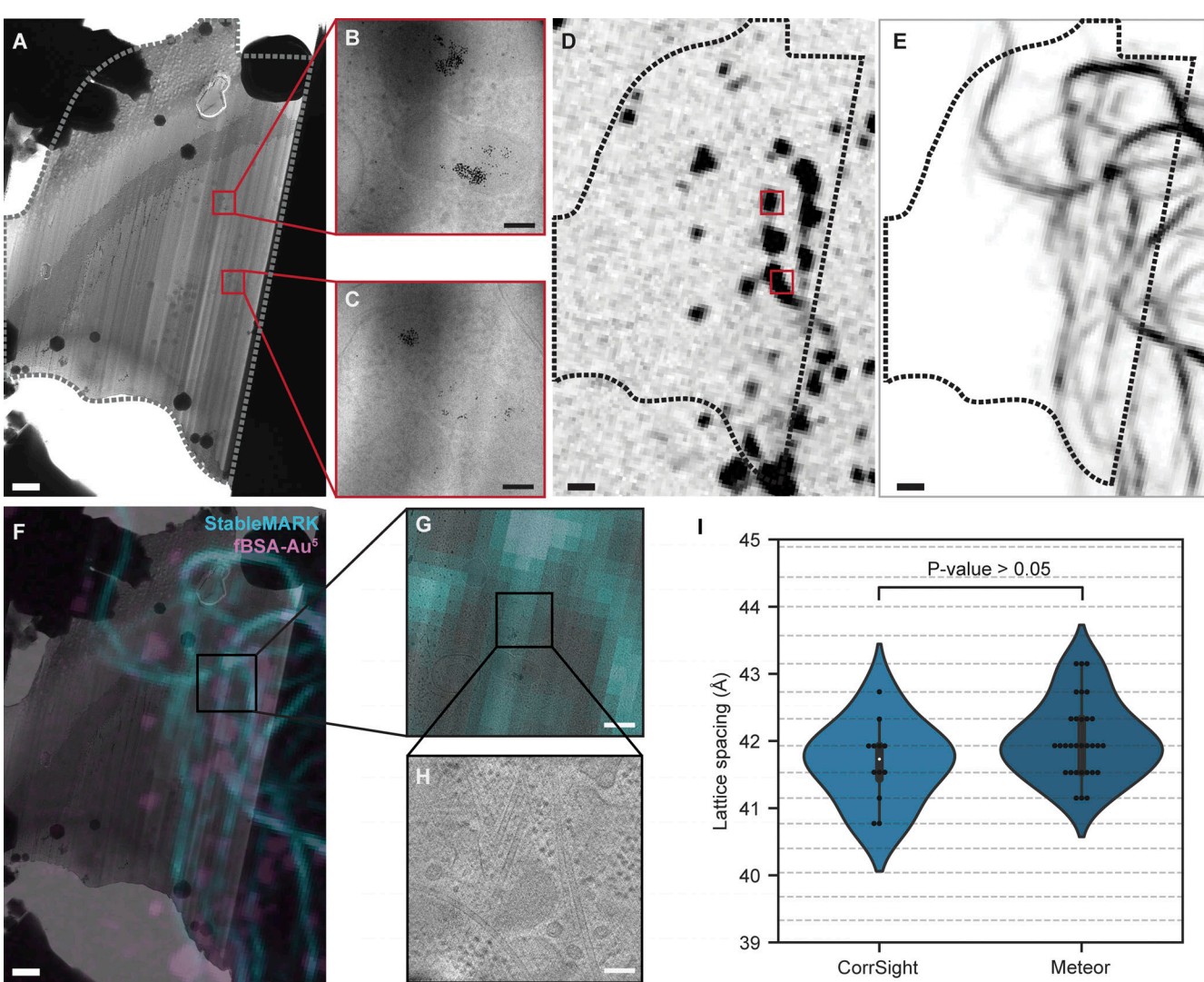

Figure S6.   **FM to TEM correlation using whole-cell FM data. (A)** TEM overview image of a lamella; dotted line indicates the outline of the lamella, red squares the location of the fBSA-Au[5] beads. **(B and C)** Zoom-in of the fBSA-Au[5] beads found both in the TEM lamella in A and in the correlated fBSA-Au[5] FM data in D. **(D)** Correlated fBSA-Au[5] FM data; red squares indicate fBSA-Au[5] location. **(E)** Correlated StableMARK-bound microtubules. **(F)** Overlay of TEM lamella and correlated FM data of both fBSA-Au[5] (pink) and StableMARK-bound microtubules (blue). **(G and H)** Two-step zoom of two microtubules overlapping with the StableMARK-bound FM data. **(I)** Violin plots showing the lattice spacing distribution of StableMARK-bound microtubules, split between microtubules correlated using FM data from CorrSight (light blue, $N$ = 12, 6 tomograms, 6 cells) or FM data from Meteor (dark blue, $N$ = 31, 19 tomograms, 17 cells). The two distributions are not significantly different (P value = 0.091, unpaired $t$ test based permutation test). Horizontal lines correspond to the discrete spatial frequency values in reciprocal space. Scale bars: 1 μm (A and D–F), 100 nm (B, C, and H), 400 nm (G).

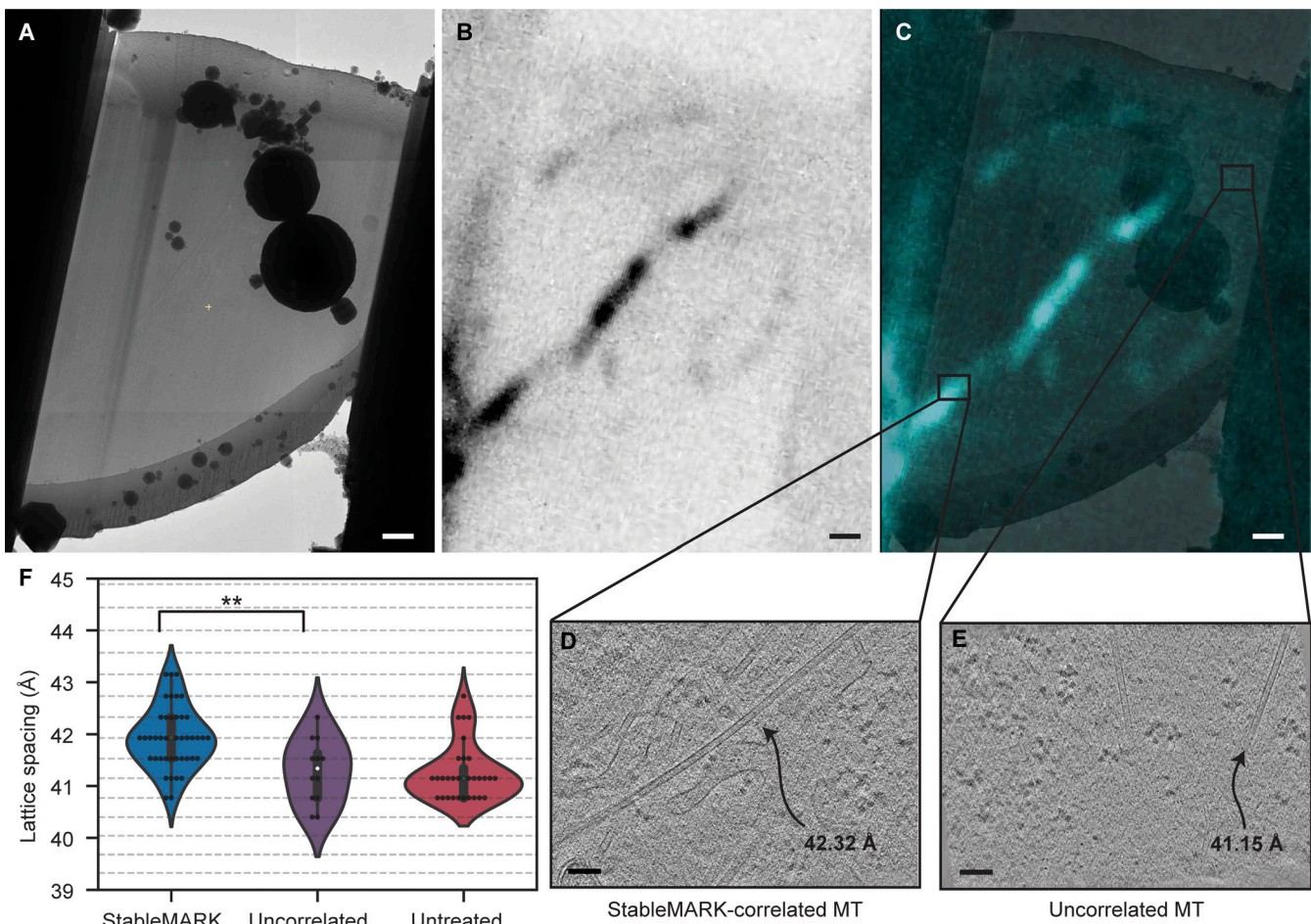

Figure S7. **Expanded and compacted microtubules can be found within one lamella. (A)** TEM overview image of a lamella. **(B)** Correlated StableMARK FM data (Meteor). **(C)** Overlay of TEM and FM data; black squares indicate the location of tomograms collected at a spot with and without StableMARK signal. **(D and E)** Tomogram slices (~4 nm thickness) of an expanded microtubule with StableMARK signal (D) and of a compacted microtubule devoid of StableMARK signal (E). **(F)** Violin plots showing the lattice spacing distribution of uncorrelated microtubules (N = 12, 10 tomograms, 8 cells), StableMARK-bound microtubules, and untreated microtubules (same data as Fig. 4 J, included for comparison). The StableMARK distribution is significantly different from the uncorrelated distribution (**P value <0.01, unpaired *t* test based permutation test). Horizontal lines correspond to the discrete spatial frequency values in reciprocal space. Scale bars: 1 µm (A–C), 100 nm (D and E).

**Provided online are Table S1 and Table S2. Table S1 lists previously reported microtubule lattice spacings with respect to nucleotide state, Taxol treatment, and microtubule-binding proteins. Table S2 shows cryo-ET data collection parameters.**

