## [Peer Review File · The Journal of Cell Biology]

StableMARK-decorated microtubules in cells have expanded lattices

Leanne de Jager, Klara Jansen, Robin Hoogebeen, Anna Akhmanova, Lukas Kapitein, Friedrich Förster, and Stuart Howes

Corresponding Author(s): Stuart Howes, Utrecht University and Friedrich Förster, Utrecht University

Review Timeline:

Submission Date:	2022-06-30
Editorial Decision:	2022-10-05
Revision Received:	2024-05-10
Editorial Decision:	2024-07-03
Revision Received:	2024-08-28
Editorial Decision:	2024-09-12
Revision Received:	2024-09-23

Monitoring Editor: Arshad Desai

Scientific Editor: Dan Simon

Transaction Report:

DOI: <https://doi.org/10.1083/jcb.202206143>

October 5, 2022

Re: JCB manuscript #202206143

Dr. Stuart Howes
Utrecht University
David de Wiedgebouw
Universiteitsweg 99
Utrecht, Utrecht 3584 CG
Netherlands

Dear Stuart,

Thank you for submitting your manuscript entitled "Increased microtubule lattice spacing correlates with selective binding of kinesin-1 in cells." Please accept our apologies for the delay in the review of your manuscript and thank you for your patience. Your paper has now been examined by three reviewers who have divergent views on the suitability of the work for publication in JCB. I agree with Reviewers 1 and 3 that the premise is indeed very interesting. However, many additional controls and new experiments would be required to convincingly demonstrate that expanded microtubules exist in cells and also that they are specifically bound by the kinesin-1 rigor mutant. Given the scope of the additional work that would be required, unfortunately our editorial decision is against publication in JCB.

Although your manuscript is intriguing, I feel that the points raised by the reviewers are more substantial than can be addressed in a typical revision period. If you wish to expedite publication of the current data, it may be best to pursue publication at another journal. However, if you feel that you can fully address the points raised by the reviewers, we would consider a revised submission. If you would like to resubmit this work to JCB, please contact the journal office to discuss an appeal of this decision or you may submit an appeal directly through our manuscript submission system. Please note that priority and novelty would be reassessed at resubmission.

Regardless of how you choose to proceed, we hope that the comments below will prove constructive as your work progresses. We would be happy to discuss the reviewer comments further once you've had a chance to consider the points raised in this letter. You can contact the journal office with any questions, cellbio@rockefeller.edu or call (212) 327-8588.

I am sorry that our decision is not more encouraging. Thank you for thinking of JCB as an appropriate place to publish your work.

Sincerely,
Rebecca

Rebecca Heald, PhD
Monitoring Editor
Journal of Cell Biology

Dan Simon, PhD
Scientific Editor
Journal of Cell Biology

Reviewer #1 (Comments to the Authors (Required)):

De Jager et al. have examined the structure of microtubules in situ using a FIB-milling + tomography approach. They extract individual microtubules out of their tomograms and subject them to layer-line analysis, which provides the lattice spacing (expanded vs. compacted). They observe both expanded and compacted lattices within a single tomogram, far away from the plus end, indicating that lattices in cells can exist in both structural states. This finding is surprising, because expanded lattices were previously thought to be confined to the GTP zone at the microtubule tip. These data indicate that the compaction following GTP hydrolysis is reversible in cells. Further, they use correlative fluorescence-cryo-ET to demonstrate that microtubules coated in a rigor kinesin have an expanded lattice.

The implication of these findings are that microtubules exist in one of 2 binary states (expanded or compacted) in addition to the gradations of post-translational modifications which can mark them. Thus, the number of microtubule "types" has been doubled. There is no doubt that this data should be published, that the work is a good fit for JCB, and that the results will be exciting to the

microtubule community. The paper is still somewhat underdeveloped, however, and I think it will benefit from another round of effort.

1. What are the controls for the layer line analysis at the core of the manuscript? I hope the authors can appreciate that the differences in reciprocal spacing shown in Fig. 1G are very small. Fig. S2D shows a similarly small difference. Even the smallest error anywhere in the pipeline could skew these results. Would it be possible, for example, to prepare a sample of microtubules *in vitro* and to measure their lattice spacing? To use GMPCPP vs. native lattices as standards for comparing to the microtubules in cells?
2. The section that compares "kinesin-bound" vs. untreated microtubules is unclear and requires reworking. For example, it's not clear to me that "kinesin-bound" and non-kinesin microtubules were analyzed from the same cell. The authors do not use consistent terminology, which makes it harder. In the results, the kinesin-bound microtubules are compared to "untargeted" microtubules, citing Fig. 3J, but then Fig. 3J talks about "untreated" microtubules, which sounds like a comparison to microtubules from cells that were not treated with taxol. What I think the manuscript needs is a side-by-side comparison from the same cell. Consider Fig. 3H, which shows a region of kinesin fluorescence, and Fig. 3I, which shows a zoom-in of the cryo data from that region. If you zoomed in on a dark region in Fig. 3H, w/out kinesin fluorescence, would you also find microtubules there? Could you then directly compare the lattice spacing of microtubules from fluorescent vs. non-fluorescent regions of the same lamellae? I'm envisioning something like Fig. S2 but for the kinesin case.
3. Speaking of Fig. S2, why isn't that in the main text? The fact that 2 microtubules of different spacings can exist side-by-side in the same cell is, in my opinion, a key draw of this paper.
4. Figure 4 is odd to me, perhaps because discrete distributions are represented as continuous, perfectly symmetrical Gaussians (the Taxol case). It also feels like filler, because it's not really presenting any new information, except perhaps to show that the kinesin expansion they observe in cells is not quite "full expansion". It's also awkward to see some data represented as distributions (top of the figure) but other data shown as single points with no error bars (bottom). Please rethink the purpose of this figure and rework accordingly.
5. The authors write about kinesin-1's "preference" for expanded lattice. The phrasing suggests that kinesin-1's affinity for expanded lattices is higher than for compacted lattices. Their data is not sufficient to demonstrate that, however. What is the evidence to show that kinesin-1 has a higher affinity for expanded lattices? Is there any evidence that either the run-length of kinesin-1, or its on-rate constant, is affected by, say, GMPCPP vs. capped GDP lattices? There is some data like that in the Shima et al. JCB paper, which allegedly showed a higher landing rate for GMPCPP microtubules vs. GDP microtubules. But I don't believe that result, because Hirokawa used 30% glycerol to stabilize the GDP microtubules. The glycerol will alter viscosity significantly, and thus diffusion coefficients, which could explain the lower landing rate. The authors need to discuss these issues and provide clarity.
6. The fact that the kinesin construct is a rigor mutant is not sufficiently emphasized, and the authors need to clarify this point. It could be the case that active kinesin-1 does not expand the lattice, for example.

Overall, I am enthusiastic about this paper and its implications for microtubule biology.

Reviewer #2 (Comments to the Authors (Required)):

In this manuscript, Jager et al. apply cryo-correlative light and electron microscopy to investigate the spacing of microtubule lattices in cells. The study is topical, given the recent increased interest in the effects of kinesin motors on microtubule lattice spacing (e.g. Peet et al. 2018; Shima et al. 2018). The authors present a two-step cryo-CLEM workflow, which allows them to investigate microtubule lattice spacings in several cellular conditions. However, the manuscript reads primarily as a methods-development attempt, and the study overall fails to provide significant new biological insight. As such, the manuscript is not appropriate for publication in JCB.

Apart from explaining their method, which takes up most of Figures 2 & 3, the authors show a total of three data sets. These include microtubule lattice spacing measurements in wt U2OS cells, taxol-treated U2OS cells, and U2OS Flp-In T-Rex cells expressing a rigor Kinesin-1 mutant. The data sets are limited (12, 4 and 6 tomograms, respectively) and information on statistical rigor and reproducibility is largely lacking (i.e. number of experimental repeats, numbers of cells analyzed, etc). The experiments seem to lack appropriate controls. For example, while the authors use correlative light microscopy to select the kinesin-1-mutant-bound microtubules, they don't seem to analyze the lattice spacings of the microtubules in the same tomograms that don't exhibit kinesin-1 binding (see Figure 3A), which should be the proper control for the experiment (rather than analysis of microtubule lattice spacing in a different cell line!). Their running title is 'Kinesin-1 selects expanded microtubule lattices', although there is no evidence as to whether the motor 'selects' expanded lattices, or rather induces the lattice expansion (as previously reported in the literature) - this study is purely correlative. Their Taxol experiment interpretation is particularly confusing - it is not clear whether the authors are using Taxol treatment (a) as a control, because of its known effects on lattice spacing, which can potentially be used as a validation of their method; or (b) as a novel result, demonstrating that taxol expands microtubule lattices in cells, but to a different extent than what was previously observed *in vitro*. In general, the authors do a fairly poor job explaining what is already known (i.e. effects of taxol reported in the literature; previous work on microtubule lattice structures in cells and what is novel in this study compared to e.g. Watanabe et al. 2020; review of the reported effects of Kinesin-1 on microtubule lattice spacing) - aspects of this show up only in their last figure (Figure 4, which also features an

undefined 'density' parameter). Overall, the manuscript appears to be a tangential study that, provided appropriate controls and statistics were to be performed, could be fully absorbed into the supplemental material of the related Kapitein lab submission (Jansen et al).

Reviewer #3 (Comments to the Authors (Required)):

The manuscript from Howes and colleagues reports the lattice spacings of microtubules in U2OS cells using cryo-CLEM. Cryo-EM of microtubules assembled in vitro revealed that the lattice spacings of microtubules composed of GDP tubulin is more compact than the one of GMPCPP microtubules and that the binding of taxol to GDP microtubules expands the lattice. Work from several groups also has shown that microtubule binding proteins can affect microtubule lattice spacing and that some motors might be able to distinguish between these two tubulin conformations that lead to the different lattice parameters. Howes et al take the analysis of microtubule lattice spacing in vivo, specifically in U2OS cells where they establish a cryo-CLEM protocol for this, made in part possible by the use of internalized gold functionalized beads to allow more precise alignment of their tomograms. Using this protocol, they show that the lattice of microtubules in cells is mostly compact, close to the GDP lattice parameters reported for in vitro assembled microtubules. Their data also shows a broader distribution of lattice parameters than expected from in vitro studies, hinting at the existence of microtubule populations within cells that have different lattices. They add then taxol to their cells and show convincingly that the lattice expands, similar again to the in vitro findings (the expansion is less pronounced than in vitro). Then they use their cryo-CLEM protocol to obtain lattice parameters of microtubules decorated with a rigor kinesin construct and here they see that these microtubules have a slightly expanded lattice. From this, they conclude that the rigor kinesin recognizes this more expanded microtubule lattice. The study is exciting, and it is, to my knowledge, the first examination of microtubule lattice parameters in cells and the effects of taxol on lattice spacing. The part of the study looking at WT microtubules and then consequences of treatment with taxol is interesting, convincing and of interest to the cell biological community. However, I am less convinced by the CLEM data with the rigor kinesin because their localization precision between the EM and the fluorescence microscopy (SD of 120 nm) makes it such that one cannot be sure what microtubule they are analyzing, especially in the region of the cell where the rigor kinesin binds where microtubules are closely spaced and some are part of bundles. I think for this aspect of the manuscript they require more data to convince me that the shift in their distribution for the lattice spacing is real.

Specific comments below:

1. It is not clear whether the lamella come from one cell for each of the conditions, or there were multiple cells for each condition from which lamella were obtained and analyzed. This needs to be clearly specified.
2. How long do the microtubule segments have to be to do the layer line analysis? Is this length requirement limiting how many MTs in their tomograms can be used to obtain lattice parameters? Are they selecting only MTs that run perpendicular to the tilt axis to minimize missing structural information?
3. In order to do the Fourier analysis, the MTs have to be straight and long enough - how was this achieved, especially when looking at the MTs around the nucleus which are more curvy and where the rigor kinesin binds.
4. They observe a broader distribution of MT lattice parameters in their control cells than for GDP microtubules in vitro - can the authors comment on the location of the MTs and whether more expanded lattices were found closer or further away from the cell periphery (see point 3 also). Adding more data here would strengthen the manuscript and possibly offer some interesting functional insights.
5. The precision of the alignment between the cryo-EM images and the fluorescence images (SD 120 nM) make it such that is not possible to be certain whether the rigor kinesin signal does not come from a microtubule that is nearby (the authors themselves acknowledge this). This is one reason for my reservation regarding the rigor kinesin data. I am aware that these experiments are tedious, but if they want to make the connection between the rigor kinesin and different lattice parameters, a larger number of microtubules will need to be analyzed to convince me that the shift in those distributions is meaningful. Also see additional control suggested in point 6 below.
6. Figure 3 compares the MT lattice spacing from the untreated cells and from the MTs that were kinesin positive from the cells expressing the rigor kinesin. I would like to see here as a control also the lattice spacings from microtubules that were not decorated by kinesin from their dox treated kinesin expressing cells.
7. The fBSA-Au5 beads used for the improved alignment are not commercial - a reference is given, but a detailed protocol of how they are made and how they were applied to cells for endocytosis needs to be given in the Methods section.

Minor

Line 52 "makes up the bulk of the microtubule core" - change the use of core, it usually implies the center of something - I think microtubule shaft is more appropriate here

We thank the reviewers for their comments and suggestions for improvements. In our revised manuscript we have:

- Increased the number of replicates for Taxol and correlated microtubules.
- Moved the focus away from kinesin-1 to address the concerns of using a rigor construct.
- Include additional *in vitro* control data for GDP, GMPCPP and Taxol lattices.

Our main findings, namely the co-existence of different microtubule lattice spacings inside a cell and the first direct evidence that stable microtubules in cells have expanded lattices, remain unchanged. Our point-by-point response follows.

Reviewer #1 (Comments to the Authors (Required)):

De Jager et al. have examined the structure of microtubules in situ using a FIB-milling + tomography approach. They extract individual microtubules out of their tomograms and subject them to layer-line analysis, which provides the lattice spacing (expanded vs. compacted). They observe both expanded and compacted lattices within a single tomogram, far away from the plus end, indicating that lattices in cells can exist in both structural states. This finding is surprising, because expanded lattices were previously thought to be confined to the GTP zone at the microtubule tip. These data indicate that the compaction following GTP hydrolysis is reversible in cells. Further, they use correlative fluorescence-cryo-ET to demonstrate that microtubules coated in a rigor kinesin have an expanded lattice.

The implication of these findings are that microtubules exist in one of 2 binary states (expanded or compacted) in addition to the gradations of post-translational modifications which can mark them. Thus, the number of microtubule "types" has been doubled. There is no doubt that this data should be published, that the work is a good fit for JCB, and that the results will be exciting to the microtubule community. The paper is still somewhat underdeveloped, however, and I think it will benefit from another round of effort.

We are pleased that the reviewer shares our excitement about the findings and agree that the paper will benefit from additional effort. Most importantly, we have now increased the dataset of correlated microtubules (from N=12 to N=43) and provide additional data for our layer line analysis using *in vitro* control experiments. The paper has shifted its focus from kinesin-1 bound microtubules to stable microtubules in general. A point-by-point response can be found below.

1. What are the controls for the layer line analysis at the core of the manuscript? I hope the authors can appreciate that the differences in reciprocal spacing shown in Fig. 1G are very small. Fig. S2D

shows a similarly small difference. Even the smallest error anywhere in the pipeline could skew these results. Would it be possible, for example, to prepare a sample of microtubules *in vitro* and to measure their lattice spacing? To use GMPCPP vs. native lattices as standards for comparing to the microtubules in cells?

As the reviewer pointed out correctly, the differences in lattice spacing are small. Thus, as suggested, we now have included control experiments where we perform our layer line analysis on *in vitro* reconstituted microtubules (Figure 1E). Using our layer line analysis, we measured the lattice spacing of microtubules *in vitro* assembled in the presence of Taxol and GMPCPP. Both lattices are expanded, and the data follow a narrow distribution. We also measured the lattice spacing of microtubules assembled without stabilizer (from GTP-tubulin) and these microtubules were predominantly compacted. These microtubules did show a broader distribution of lattice spacings, which is probably due to its dynamic nature and due to measuring the signal only within a single microtubule, as is novel here, and not across microtubules as is common in the literature where only the dominant/majority value is determined during the averaging. The values of these 3 datasets are in line with what has been reported in the literature and therefore confirm the reliability of our layer line analysis for individual microtubules under the dose regimes utilized here.

2. The section that compares "kinesin-bound" vs. untreated microtubules is unclear and requires reworking. For example, it's not clear to me that "kinesin-bound" and non-kinesin microtubules were analyzed from the same cell. The authors do not use consistent terminology, which makes it harder. In the results, the kinesin-bound microtubules are compared to "untargeted" microtubules, citing Fig. 3J, but then Fig. 3J talks about "untreated" microtubules, which sounds like a comparison to microtubules from cells that were not treated with taxol. What I think the manuscript needs is a side-by-side comparison from the same cell. Consider Fig. 3H, which shows a region of kinesin fluorescence, and Fig. 3I, which shows a zoom-in of the cryo data from that region. If you zoomed in on a dark region in Fig. 3H, w/out kinesin fluorescence, would you also find microtubules there? Could you then directly compare the lattice spacing of microtubules from fluorescent vs. non-fluorescent regions of the same lamellae? I'm envisioning something like Fig. S2 but for the kinesin case.

We agree that inconsistencies may have been confusing and have now taken extra care at keeping our terminology consistent by always using the term untreated microtubules. The untreated dataset is obtained in U2OS cells and the correlated stable microtubule data set is obtained in U2OS Flp-In T-Rex cells. Although not identical, these two cell types are highly similar, and we do not expect their microtubule networks to be materially different. Furthermore, the compacted lattice spacing is well

established in literature, is not considered controversial, and we do observe the compacted lattice spacing in U2OS Flp-In T-Rex cells (see Supplemental Figure 6), consistent with all *in vitro* observations. Finally, we did measure the lattice spacing of a correlated microtubule by transiently transfecting WT U2OS cells with the StableMARK construct before switching to a stable cell line to increase the throughput (see included Figure R1). The lattice spacing for this microtubule was 41.9 Å, consistent with that observed in the U2OS Flp-In T-Rex cells. Therefore, all our data support the conclusion that stable microtubules have an expanded lattice compared to the compacted state.

We would like to reiterate that our correlative workflow is useful to positively localize StableMARK but cannot be used to definitively exclude the presence of StableMARK, as sparsely decorated microtubules might appear undecorated/without signal, due to the limited sensitivity of our cryo-light microscopes and the deliberately low expression levels chosen to minimize the introduction of over-expression artifacts. We also do not control for binding of unlabeled kinesins.

Figure R1: The correlated microtubule from a U2OS cell transiently transfected with StableMARK. (A) Snapshot of the relevant lamella, red outline showing where the tilt series was collected and white arrow indicating where the fBSA-Au⁵ beads that confirmed correct correlation were found. (B) Snapshot of the tomogram with the correlated microtubule (90 nm thickness). (C) Line profile plot of the microtubule shown in (B), arrow pointing out the location of the layer line peak, indicating that it is expanded. Scale bars: 200 nm (A) and 100 nm (B).

3. Speaking of Fig. S2, why isn't that in the main text? The fact that 2 microtubules of different spacings can exist side-by-side in the same cell is, in my opinion, a key draw of this paper.

Although we did not observe the existence of different lattice spacing side-by-side regularly, we do agree that it is a very relevant phenomena and therefore have now merged this figure with Figure 1 (see Figure 1A-D).

4. Figure 4 is odd to me, perhaps because discrete distributions are represented as continuous, perfectly symmetrical Gaussians (the Taxol case). It also feels like filler, because it's not really presenting any new information, except perhaps to show that the kinesin expansion they observe in

cells is not quite "full expansion". It's also awkward to see some data represented as distributions (top of the figure) but other data shown as single points with no error bars (bottom). Please rethink the purpose of this figure and rework accordingly.

We see how this figure could be awkward for some readers. However, we do find it relevant to position our data on *in situ* lattice spacing in the context of previously reported lattice spacings. Therefore, we have redesigned the figure (now Supplemental Figure 1), which now shows a summary of reported spacings for microtubules bound by different small molecules (e.g., Taxol and GMPCPP) or by proteins (like Tau and MAP7). This allows the reader to more easily compare our findings with previously reported lattice spacings.

5. The authors write about kinesin-1's "preference" for expanded lattice. The phrasing suggests that kinesin-1's affinity for expanded lattices is higher than for compacted lattices. Their data is not sufficient to demonstrate that, however. What is the evidence to show that kinesin-1 has a higher affinity for expanded lattices? Is there any evidence that either the run-length of kinesin-1, or its on-rate constant, is affected by, say, GMPCPP vs. capped GDP lattices? There is some data like that in the Shima et al. JCB paper, which allegedly showed a higher landing rate for GMPCPP microtubules vs. GDP microtubules. But I don't believe that result, because Hirokawa used 30% glycerol to stabilize the GDP microtubules. The glycerol will alter viscosity significantly, and thus diffusion coefficients, which could explain the lower landing rate. The authors need to discuss these issues and provide clarity.

The concerns of the reviewer are valid, and we agree our initial attempts to draw conclusions between the rigor StableMARK and kinesin-1 were not appropriately justified by the data included. Now that StableMARK is a published reagent it is less central to this article and we have shifted our focus from the preference of kinesin-1 for expanded microtubules towards the lattice spacing of stable microtubules in general.

6. The fact that the kinesin construct is a rigor mutant is not sufficiently emphasized, and the authors need to clarify this point. It could be the case that active kinesin-1 does not expand the lattice, for example.

To emphasize the use of the rigor-mutant more, we now mention it multiple times, namely where the correlative workflow and experiment design is introduced (line 131) and in the results summary (line 192). The fact that StableMARK is a rigor construct is repeated in the materials and methods at line 228 and 243 as well.

Overall, I am enthusiastic about this paper and its implications for microtubule biology.

Reviewer #2 (Comments to the Authors (Required)):

In this manuscript, Jager et al. apply cryo-correlative light and electron microscopy to investigate the spacing of microtubule lattices in cells. The study is topical, given the recent increased interest in the effects of kinesin motors on microtubule lattice spacing (e.g. Peet et al. 2018; Shima et al. 2018). The authors present a two-step cryo-CLEM workflow, which allows them to investigate microtubule lattice spacings in several cellular conditions. However, the manuscript reads primarily as a methods-development attempt, and the study overall fails to provide significant new biological insight. As such, the manuscript is not appropriate for publication in JCB.

The methods implemented to carry out this study, while conceptually described previously and utilized in other systems, represent a significant resource investment that is prohibitive for many research groups and partially explains why it has not been undertaken until now. This is also why the initial number of replicates (n) was low. We agree the methods potentially received too much emphasis in the original manuscript. However, we disagree that there is no significant biological insight, which is also reflected by the context of the manuscript's citations to date.

The increased number of replicates required was an important point raised by multiple reviewers that we have now addressed. The original conclusions have not changed and the implications of different lattice spacing within the shaft of the microtubule can be considered highly interesting. Substantial literature exists on the lattice space differences of microtubules in vitro. The fact that we now, for the first time, confirm that there is also diversity inside cells opens up a whole new avenue of research. For example, the binding dynamics of microtubule proteins, how microtubule subset are defined, or the regulation of cellular processes like cargo transport and cell division might now be studied from the perspective of variable lattice spacing.

The interest from the community in our work and lattice spacing in general is clear to us, as our preprint has already been cited in various preprints and papers. For example, Shen and Ori-McKenney studied how the MAP code responds to macromolecular crowding and hypothesized that lattice expansion for MAP7 is relevant in this process (see doi: 10.1101/2023.06.14.544846, (Shen and Ori-McKenney, 2023)). Furthermore, Yue et al. have found a link between lattice expansion and the deetyrosination activity of VASH1/SVBP. This became even more relevant since we showed that lattice variability exists within cells (see doi: 10.1016/j.cub.2023.07.062 (Yue et al., 2023)). Finally, Siahaan et

al. studied the role of lattice spacing in tau envelop formation, using Taxol to induce an expanded state inside cells as part of their experiments (see doi: 10.1038/s41589-022-01096-2 (McKenney, 2022; Siahhan et al., 2022)). These papers, together with feedback at conferences, give us the confidence that this work will be well received and useful to share via publication.

Apart from explaining their method, which takes up most of Figures 2 & 3, the authors show a total of three data sets. These include microtubule lattice spacing measurements in wt U2OS cells, taxol-treated U2OS cells, and U2OS Flp-In T-Rex cells expressing a rigor Kinesin-1 mutant. The data sets are limited (12, 4 and 6 tomograms, respectively) and information on statistical rigor and reproducibility is largely lacking (i.e. number of experimental repeats, numbers of cells analyzed, etc).

We agree with the reviewer that the datasets were limited and that reproducibility numbers were less clear. We have now increased our Taxol and correlative StableMARK datasets substantially, from N=12 to N=30 and N=12 to N=43 microtubules for Taxol and StableMARK respectively, see figures 2 and 3. We have now stated per dataset in the figure legends (e.g., line 623, 633, 661 and 717) how many cells were analysed as well.

The experiments seem to lack appropriate controls. For example, while the authors use correlative light microscopy to select the kinesin-1-mutant-bound microtubules, they don't seem to analyze the lattice spacings of the microtubules in the same tomograms that don't exhibit kinesin-1 binding (see Figure 3A), which should be the proper control for the experiment (rather than analysis of microtubule lattice spacing in a different cell line!).

The reviewer points out correctly that the untreated dataset is obtained in U2OS cells and the correlated StableMARK microtubule dataset is obtained in U2OS Flp-In T-Rex cells. Although not the same, these two cell types are highly similar and we do not expect their microtubule networks to be materially different. Furthermore, the compacted lattice spacing is well established in literature, and it is not controversial in the field. We do observe the compacted lattice spacing in U2OS Flp-In T-Rex cells (see Supplemental Figure 6), consistent with all *in vitro* observations. Finally, we did measure the lattice spacing of a correlated microtubule by transiently transfecting WT U2OS cells with the StableMARK construct before switching to a stable cell line to increase the throughput (see included Figure R1 in response to reviewer #1). The lattice spacing for this microtubule was 41.9 Å, consistent with that observed in the U2OS Flp-In T-Rex cells. Therefore, all our data support the conclusion that stable microtubules have an expanded lattice compared to the compacted state.

We would like to reiterate that our correlative workflow is useful to positively localize StableMARK but cannot be used to definitively exclude the presence of StableMARK, as sparsely decorated microtubules might appear undecorated/without signal, due to the limited sensitivity of the cryo-light microscopes, and the deliberately low expression levels chosen to minimize the introduction of over-expression artifacts. We also do not control for binding of unlabeled kinesins.

Their running title is 'Kinesin-1 selects expanded microtubule lattices', although there is no evidence as to whether the motor 'selects' expanded lattices, or rather induces the lattice expansion (as previously reported in the literature) - this study is purely correlative.

We agree that our data does not directly support the conclusion that kinesin-1 preferentially binds to expanded lattices. We have modified the title to “Stable microtubules in cells have expanded lattices” and, in light of the publication of StableMARK as a live-cell reagent, we have shifted the focus to emphasize here that stable microtubules have an expanded state.

Their Taxol experiment interpretation is particularly confusing - it is not clear whether the authors are using Taxol treatment (a) as a control, because of its known effects on lattice spacing, which can potentially be used as a validation of their method; or (b) as a novel result, demonstrating that taxol expands microtubule lattices in cells, but to a different extent than what was previously observed *in vitro*.

We agree with the reviewer that the double role of the Taxol data as both a control and a finding in the original manuscript was confusing. To clarify this, we now clearly state that the expanded state found inside cells upon Taxol treatment is a novel finding. With better statistics and a more rigorous evaluation of our processing of data from single microtubules (Supplemental Figure 2B), we are more certain that this is a novel lattice state, and clearly state this. Furthermore, during our revision period, other groups have also used it as a way to modulate the microtubule lattice inside cells (Yue et al., 2023) and (Siahaan et al., 2022). The novel lattice spacing also can point towards an additional explanation for its mechanism of action as an anti-cancer drug, which we believe is an important point to follow-up in future research.

Additionally, we have now included control experiments where we measured the lattice spacing of microtubules assembled *in vitro* in the presence of Taxol and GMPCPP using our layer line analysis. Both lattices are expanded and the data follow a narrow distribution. We also measured the lattice spacing of microtubules assembled without stabilizer (from GTP-tubulin) and these microtubules were predominantly compacted (Figure 1E, Supplemental Figure 3). These microtubules did show a broader distribution of lattice spacings, which is probably due to its more dynamic nature and due to measuring

the signal only within a single microtubule, as is novel here, and not across microtubules as is common in the literature where only the dominant/majority value is determined during the averaging. The values of these 3 datasets are in line with what is known in literature and therefore confirm the reliability of our layer line analysis.

In general, the authors do a fairly poor job explaining what is already known (i.e. effects of taxol reported in the literature; previous work on microtubule lattice structures in cells and what is novel in this study compared to e.g. Watanabe et al. 2020; review of the reported effects of Kinesin-1 on microtubule lattice spacing) - aspects of this show up only in their last figure (Figure 4, which also features an undefined 'density' parameter)

We agree with the reviewer that it is important to compare our findings to what is known in the literature. Therefore, we have redesigned the figure (now Supplemental Figure 1) which now shows a summary of reported spacings for microtubules bound by different small molecules (e.g., Taxol and GMPCPP) or by proteins (like Tau and MAP7). This allows the reader to more easily compare our findings with previously reported lattice spacings.

Overall, the manuscript appears to be a tangential study that, provided appropriate controls and statistics were to be performed, could be fully absorbed into the supplemental material of the related Kapitein lab submission (Jansen et al).

While these two papers do use similar reagents and investigate similar features of the microtubule cytoskeleton, we do not see them as part of the same study. Given the relevance and implications of the presence of variability in lattice spacing within cells for the microtubule community, it would be a loss if this information would end up in the supplements of any given paper and potentially get missed by the community. Furthermore, we believe this would also not do justice to the large amount of resources and time that has been invested in setting up the correlative workflow and in acquiring the data in this manuscript.

Reviewer #3 (Comments to the Authors (Required)):

The manuscript from Howes and colleagues reports the lattice spacings of microtubules in U2OS cells using cryo-CLEM. Cryo-EM of microtubules assembled in vitro revealed that the lattice spacings of microtubules composed of GDP tubulin is more compact than the one of GMPCPP microtubules and that the binding of taxol to GDP microtubules expands the lattice. Work from several groups also has shown that microtubule binding proteins can affect microtubule lattice spacing and that some motors might be able to distinguish between these two tubulin conformations that lead to the different lattice

parameters. Howes et al take the analysis of microtubule lattice spacing in vivo, specifically in U2OS cells where they establish a cryo-CLEM protocol for this, made in part possible by the use of internalized gold functionalized beads to allow more precise alignment of their tomograms. Using this protocol, they show that the lattice of microtubules in cells is mostly compact, close to the GDP lattice parameters reported for in vitro assembled microtubules. Their data also shows a broader distribution of lattice parameters than expected from in vitro studies, hinting at the existence of microtubule populations within cells that have different lattices. They add then taxol to their cells and show convincingly that the lattice expands, similar again to the in vitro findings (the expansion is less pronounced than in vitro). Then they use their cryo-CLEM protocol to obtain lattice parameters of microtubules decorated with a rigor kinesin construct and here they see that these microtubules have a slightly expanded lattice. From this, they conclude that the rigor kinesin recognizes this more expanded microtubule lattice. The study is exciting, and it is, to my knowledge, the first examination of microtubule lattice parameters in cells and the effects of taxol on lattice spacing. The part of the study looking at WT microtubules and then consequences of treatment with taxol is interesting, convincing and of interest to the cell biological community. However, I am less convinced by the CLEM data with the rigor kinesin because their localization precision between the EM and the fluorescence microscopy (SD of 120 nm) makes it such that one cannot be sure what microtubule they are analyzing, especially in the region of the cell where the rigor kinesin binds where microtubules are closely spaced and some are part of bundles. I think for this aspect of the manuscript they require more data to convince me that the shift in their distribution for the lattice spacing is real.

Specific comments below:

1. It is not clear whether the lamella come from one cell for each of the conditions, or there were multiple cells for each condition from which lamella were obtained and analyzed. This needs to be clearly specified.

We agree with the reviewer that this should be made clearer. We have now stated per dataset in the figure legends (e.g., line 623, 633, 661 and 717) how many cells were analysed as well. We have analysed 7 cells for the untreated condition, 5 cells for the Taxol condition and 23 cells for the correlated/StableMARK condition.

2. How long do the microtubule segments have to be to do the layer line analysis? Is this length requirement limiting how many MTs in their tomograms can be used to obtain lattice parameters?

Are they selecting only MTs that run perpendicular to the tilt axis to minimize missing structural information?

We did not determine the minimal segment/box size required to confidently assign lattice spacings. Other factors, such as tomogram thickness and lamella stability, played a more significant role making it challenging to determine a reliable value. Using the *in vitro* data to estimate a minimal segment length would also not be useful as the background signal for micrographs of purified components is much lower. The strength of the signal does indeed also depend on the orientation of the microtubule with respect to the tilt axis, but only across a very narrow range is the signal lost since the microtubule lattice spacing is such a dominant feature, so overall the orientation is not limiting.

We also now include the effect of different box sizes on the SNR of the peaks (see Supplemental Figure 2B) across a range that balances what is computationally feasible against insufficient tubulin copy number to obtain sufficient, reliable signal. We show there that the SNR does not differ substantially between different box sizes. Since smaller box sizes are computationally much more efficient than bigger box sizes while having a comparable SNR, we have used a box size of 1030 pixels, corresponding to a segment length of 224 nm (minimally 26 heterodimers head-to-tail) to ensure the results are reliable, with the limitation that within that segment any differences are averaged together. The segment length is now given in the methods (lines 396-7).

3. In order to do the Fourier analysis, the MTs have to be straight and long enough - how was this achieved, especially when looking at the MTs around the nucleus which are more curvy and where the rigor kinesin binds.

Yes, the Fourier analysis does work best when the microtubules are straight. However, even in cells the microtubule bending mostly happens across micrometers while the field of view we are using is approximately 1 micrometer and the segments used are smaller (224 nm), meaning this curvature is not a major issue. Furthermore, the blurring of the layer lines due to curvature is more prominent at the ends of the layer lines, and by taking the profile plot, the blur contributes to the background, but does not change the position of the peak. The few microtubules that were highly bent even within the shorter segments, were excluded from analysis.

4. They observe a broader distribution of MT lattice parameters in their control cells than for GDP microtubules *in vitro* - can the authors comment on the location of the MTs and whether more expanded lattices were found closer or further away from the cell periphery (see point 3 also). Adding

more data here would strengthen the manuscript and possibly offer some interesting functional insights.

We do expect that expanded lattices will be found more frequently around the nuclear periphery while compacted lattices will be more prevalent nearer to the periphery of the cell. However, the SEM images taken during milling do not allow for confident localization of the cell edge and nucleus. While we could include this hypothesis in the discussion, because we did not rigorously sample the various regions of a cell due to the significant extra work this would entail (additional correlative steps for nuclear and plasma membrane stains), we feel it goes beyond the scope of our data. This is definitely interesting for future research.

5. The precision of the alignment between the cryo-EM images and the fluorescence images (SD 120 nm) make it such that it is not possible to be certain whether the rigor kinesin signal does not come from a microtubule that is nearby (the authors themselves acknowledge this). This is one reason for my reservation regarding the rigor kinesin data. I am aware that these experiments are tedious, but if they want to make the connection between the rigor kinesin and different lattice parameters, a larger number of microtubules will need to be analyzed to convince me that the shift in those distributions is meaningful. Also see additional control suggested in point 6 below.

The reviewer indeed points out correctly that the difference in resolution between cryo-LM and cryo-EM data makes correlation challenging. To improve the reliability of our findings we repeated our experiments to expand our dataset and have now analyzed 43 correlated microtubules instead of 12 (see Figure 4). With this bigger dataset we draw the same conclusion that correlated stable microtubules are expanded compared to the established compacted lattice.

6. Figure 3 compares the MT lattice spacing from the untreated cells and from the MTs that were kinesin positive from the cells expressing the rigor kinesin. I would like to see here as a control also the lattice spacings from microtubules that were not decorated by kinesin from their dox treated kinesin expressing cells.

The untreated dataset is obtained in U2OS cells and the correlated stable microtubule data set is obtained in U2OS Flp-In T-Rex cells. Although not identical, these two cell types are highly similar and we do not expect their microtubule networks to be materially different. Furthermore, the compacted lattice spacing is extremely well established in the literature, is not considered controversial, and we do observe the compacted lattice spacing in U2OS Flp-In T-Rex cells (see Supplemental Figure 6),

consistent with all *in vitro* observations. Finally, we did measure the lattice spacing of a correlated microtubule by transiently transfecting WT U2OS cells with the StableMARK construct before switching to a stable cell line to increase the throughput (see included Figure R1 in response to reviewer #1). The lattice spacing for this microtubule was 41.9 Å, consistent with that observed in the U2OS Flp-In T-Rex cells. Therefore, all our data supports the conclusion that stable microtubules have an expanded lattice compared to the compacted state.

We would like to reiterate that our correlative workflow is useful to positively localize StableMARK but cannot be used to definitively exclude the presence of the StableMARK, as sparsely decorated microtubules might appear undecorated/without signal, due to the limited sensitivity of our cryo-light microscopes, and the deliberately low expression levels chosen to minimize the introduction of over-expression artifacts. We also do not control for binding of unlabeled kinesins.

7. The fBSA-Au5 beads used for the improved alignment are not commercial - a reference is given, but a detailed protocol of how they are made and how they were applied to cells for endocytosis needs to be given in the Methods section.

Since submission of the first version of this manuscript, the fBSA-Au5 beads have become commercially available. We mention in line 281 how we used the beads and in the list of reagents where they can be purchased.

Minor

Line 52 "makes up the bulk of the microtubule core" - change the use of core, it usually implies the center of something - I think microtubule **shaft** is more appropriate here.

We agree with the reviewer that shaft is a better term, and we now use the word shaft instead of core throughout the manuscript.

References (relevant for reviewer 2):

- McKenny, R. J. (2022) Alzheimer's disease-implicated protein tau puts the squeeze on microtubules. *Nature Chemical Biology*, 18(11), 1172-1173. 10.1038/s41589-022-01097-1.
- Shen, Y. & Ori-McKenney, K. M. (2023) Macromolecular Crowding Tailors the Microtubule Cytoskeleton Through Tubulin Modifications and Microtubule-Associated Proteins. *bioRxiv*, 2023.06.14.544846. 10.1101/2023.06.14.544846.
- Siahaan, V., Tan, R., Humhalova, T., Libusova, L., Lacey, S. E., Tan, T., Dacy, M., Ori-McKenney, K. M., McKenney, R. J., Braun, M. & Lansky, Z. (2022) Microtubule lattice spacing governs cohesive

envelope formation of tau family proteins. *Nature Chemical Biology*, 18(11), 1224-1235. 10.1038/s41589-022-01096-2.

Yue, Y., Hotta, T., Higaki, T., Verhey, K. J. & Ohi, R. (2023) Microtubule detyrosination by VASH1/SVBP is regulated by the conformational state of tubulin in the lattice. *Current Biology*, 33(19), 4111-4123.e7. 10.1016/j.cub.2023.07.062.

July 3, 2024

Re: JCB manuscript #202206143R-A

Dr. Stuart Howes
Utrecht University
David de Wiedgebouw
Universiteitsweg 99
Utrecht, Utrecht 3584 CG
Netherlands

Dear Dr. Howes,

Thank you for submitting your revised manuscript entitled "Stable microtubules in cells have expanded lattices." The manuscript has been seen by the three original reviewers whose full comments are appended below.

You will see that while Reviewers #1&2 feel that the study is now suitable for JCB, pending text revisions to improve accuracy and accessibility, Reviewer #3 expresses concerns about high taxol concentration and inclusion of biotinylated and fluorescently labeled tubulins in the new in vitro assays. Reviewer #3 also states that the methods section is still missing important details about the image processing used to determine differences in microtubule lattice spacing. We agree that these are important issues that we would like to give you a chance to address prior to reaching a final decision on publication in the journal.

Our general policy is that papers are considered through only one major revision cycle, but in this case we are open to an additional final round of revisions. If you choose to resubmit, a revised manuscript will have to thoroughly address the technical issues raised by Reviewer #3 either with new data or compelling arguments justifying the experimental conditions. Please also include all requested method details and revise the text to improve accuracy as well as accessibility for non-specialists as mentioned by Reviewer #1. Please note that a revised manuscript will not be considered unless all of the comments are sufficiently addressed.

Please submit the final revision along with a cover letter that includes a point by point response to the remaining reviewer comments.

Thank you for this interesting contribution to Journal of Cell Biology. You can contact me or the scientific editor listed below at the journal office with any questions at cellbio@rockefeller.edu.

Sincerely,

Arshad Desai, PhD
Monitoring Editor
Journal of Cell Biology

Dan Simon, PhD
Scientific Editor
Journal of Cell Biology

Reviewer #1 (Comments to the Authors (Required)):

This revised manuscript is significantly improved over the initial submission. The extent of data (replicates) has increased in such a way as to solidify the findings, and I believe replicate numbers are clearly sufficient, particularly given the technological feats involved. I believe the paper should be accepted with only text-based tweaks.

The title was changed to avoid reference to kinesin-1 and its affinity, but I find the new title too broad. The authors have shown that StableMARK-decorated microtubules are expanded, but it's an extrapolation to move from "StableMARK-decorated" to "Stable". Other types of stable microtubules are likely compacted, e.g. those coated in tau protein. I trust that the authors can find a happy medium between their initial title and their new one.

Obviously the paper remains somewhat technical in nature, as Reviewer 2 noted in the first round. I don't see an intrinsic problem with that, except that some of the technical writing is hard to follow, because the level of description is uneven. For example, the authors include precise SEM localization errors in the main text for the second workflow, but not the first, and the

comparison to Arnold 2016 is meaningless to the non-specialist without also stating numbers here. The subsequent explanation for the "slight increase in accuracy" is vague. I encourage the authors to think one more time about (1) what a non-specialist needs to know to understand the main benefits/pitfalls/caveats of the two cryo-CLEM workflows and to validate the accuracy, (2) what a specialist needs to know to repeat the experiments or implement your analysis pipelines. Separate these two types of text into the main text versus the methods / supplement to the extent possible.

Otherwise, I look forward to seeing this paper in print.

Reviewer #2 (Comments to the Authors (Required)):

In the revised manuscript, the authors have significantly increased the number of data samples, allowing for more adequate statistics. Furthermore, they have reorganized and reframed the manuscript to provide clearer and more rigorous presentation and interpretation of the results.

Reviewer #3 (Comments to the Authors (Required)):

Using cryo-correlative light and electron microscopy, de Jager and colleagues investigated microtubule lattice geometry in cells. They report that cells primarily have compacted microtubule lattices with lattice spacing similar to that of dynamic microtubules assembled *in vitro*. On the other hand, microtubules stabilized with Taxol or decorated with kinesin-1 rigor (StableMARK) have expanded lattices *in vivo*. Overall, the findings are narrow in scope and do not seem suitable for publication in the Journal of Cell Biology which prides itself on rigorous and comprehensive work. Regardless of where this work will be published, experimental revisions and more detailed methods are needed.

In vitro effects of Taxol on the microtubule lattice described here, and previously, were documented using high concentrations of Taxol. The authors polymerized microtubules *in vitro* using equimolar concentration of tubulin to Taxol (~20 μ M). Cells were treated with 1 μ M Taxol in their experiments, and published work has shown that under these treatment conditions, intra-cellular concentrations of Taxol are in the range of tens to hundreds of nM at most, but definitely not anything close to the concentrations used *in vitro* by the authors. Also, U2OS cells pump out taxol with the help of P-glycoprotein, so it is not clear at all how to compare the *in vitro* and *in situ* results the way the experiments were designed, and what the differences they see are due to. Also, strangely and of concern, the authors used 12% HiLyte Fluor and 18% biotinylated tubulin to assemble GMPCPP microtubules, which were then flash-frozen at -80{degree sign}C. High concentrations of labeled tubulin in the microtubule lattice and freezing/thawing impact microtubule dynamics, and therefore can have an effect on lattice spacing. Taxol-stabilized microtubules used in these experiments were also labeled with high rhodamine and biotin-tubulin. These *in vitro* experiments seem to have been done in a haphazard way. What is the purpose of using biotinylated and fluorescently labeled tubulin for these EM experiments (they certainly do not exist in cells)?

The manuscript describes small changes in the microtubule lattice using precise image processing protocols. Therefore, these protocols need to be described in greater detail than what is provided now. For example, the authors need to be more detailed in the Methods section when describing how they performed subtomogram averaging to estimate protofilament number in microtubules and to determine helical symmetry parameters. What volume did they use as an initial reference? If the initial reference was a 13-stranded microtubule, how did they avoid reference bias? The authors should reconstruct individual microtubules (e.g., using approaches described in Foster, H.E. et al., 2022, JCB) to double-check protofilament number distribution in Taxol-treated and control cells. Additionally, how did the authors perform helical averaging to find helical parameters? Did they obtain initial helical parameters from a C1 reconstruction and then refined those parameters iteratively to obtain the final reported values?

Fig.4 Panel J: The authors have tomography data collected on lamella areas with low or no StableMARK signal (Fig. S6). Microtubule lattice spacing distribution on those areas should be added to the graph.

A summary Supplementary Table listing cryo-electron tomography datasets and collection parameters for each of the datasets is needed.

We thank the reviewers and editors for their comments and suggestions for improvements. We have included two versions of the revised manuscript, one where we have highlighted all the changes and one with standard formatting. In the point-by-point responses below we give the line numbers for the most relevant changes.

Reviewer #1 (Comments to the Authors (Required)):

This revised manuscript is significantly improved over the initial submission. The extent of data (replicates) has increased in such a way as to solidify the findings, and I believe replicate numbers are clearly sufficient, particularly given the technological feats involved. I believe the paper should be accepted with only text-based tweaks.

The title was changed to avoid reference to kinesin-1 and its affinity, but I find the new title too broad. The authors have shown that StableMARK-decorated microtubules are expanded, but it's an extrapolation to move from "StableMARK-decorated" to "Stable". Other types of stable microtubules are likely compacted, e.g. those coated in tau protein. I trust that the authors can find a happy medium between their initial title and their new one.

We thank the reviewer for this insightful comment and changed the title from "Stable microtubules in cells have expanded lattices" to "StableMARK-decorated microtubules in cells have expanded lattices".

Obviously the paper remains somewhat technical in nature, as Reviewer 2 noted in the first round. I don't see an intrinsic problem with that, except that some of the technical writing is hard to follow, because the level of description is uneven. For example, the authors include precise SEM localization errors in the main text for the second workflow, but not the first, and the comparison to Arnold 2016 is meaningless to the non-specialist without also stating numbers here. The subsequent explanation for the "slight increase in accuracy" is vague. I encourage the authors to think one more time about (1) what a non-specialist needs to know to understand the main benefits/pitfalls/caveats of the two cryo-CLEM workflows and to validate the accuracy, (2) what a specialist needs to know to repeat the experiments or implement your analysis pipelines. Separate these two types of text into the main text versus the methods / supplement to the extent possible.

Otherwise, I look forward to seeing this paper in print.

We thank the reviewer for the helpful comments about how best to frame and separate the information. We have made changes in phrasing and vocabulary to clarify things, and we have added additional explanation in lines **82-84**, **145-151**, **158**, **162-166** and **174** in the main text and in lines **290-292**, **347**, **357**, **424**, **432-443** and **447-452** in the Materials and Methods section. This should make it better to read for a non-specialist and easier to repeat for a specialist.

Furthermore, we have now made the localization estimations between the two workflows consistent by doing the leave-one-out measurement for the Meteor as was done for the CorrSight workflow. For the Meteor workflow, a slightly larger error was calculated (**line 162**, Figure 3E). This is likely due to the fact that the smaller field of view of the Meteor resulted in fewer beads that could be used for the measurement (4-8 beads instead of 7-9 beads for the CorrSight workflow). Nevertheless, this is still well below the cryo-light microscope resolution and better than error values reported by Arnold et al. We have now specifically stated the calculated error determined by Arnold et al. in **line 164** to clarify our comparison for the reader.

Reviewer #2 (Comments to the Authors (Required)):

In the revised manuscript, the authors have significantly increased the number of data samples, allowing for more adequate statistics. Furthermore, they have reorganized and reframed the manuscript to provide clearer and more rigorous presentation and interpretation of the results.

We thank the reviewer for their positive comments.

Reviewer #3 (Comments to the Authors (Required)):

Using cryo-correlative light and electron microscopy, de Jager and colleagues investigated microtubule lattice geometry in cells. They report that cells primarily have compacted microtubule lattices with lattice spacing similar to that of dynamic microtubules assembled in vitro. On the other hand, microtubules stabilized with Taxol or decorated with kinesin-1 rigor (StableMARK) have expanded lattices in vivo. Overall, the findings are narrow in scope and do not seem suitable for publication in the Journal of Cell Biology which prides itself on rigorous and comprehensive work. Regardless of where this work will be published, experimental revisions and more detailed methods are needed.

In vitro effects of Taxol on the microtubule lattice described here, and previously, were documented using high concentrations of Taxol. The authors polymerized microtubules in vitro using equimolar concentration of tubulin to Taxol (~20 μ M). Cells were treated with 1 μ M Taxol in their experiments, and published work has shown that under these treatment conditions, intracellular concentrations of Taxol are in the range of tens to hundreds of nM at most, but definitely not anything close to the concentrations used in vitro by the authors. Also, U2OS cells pump out taxol with the help of P-glycoprotein, so it is not clear at all how to compare the in vitro and in situ results the way the experiments were designed, and what the differences they see are due to. Also, strangely and of concern, the authors used 12% HiLyte Fluor and 18% biotinylated tubulin to assemble GMPCPP microtubules, which were then flash-frozen at -80 °C. High concentrations of labeled tubulin in the microtubule lattice and freezing/thawing impact microtubule dynamics, and therefore can have an effect on lattice spacing. Taxol-stabilized microtubules used in these experiments were also labeled with high rhodamine and biotin-tubulin. These in vitro experiments seem to have been done in a haphazard way. What is the purpose of using biotinylated and fluorescently labeled tubulin for these EM experiments (they certainly do not exist in cells)?

We thank the reviewer for the feedback and comments to improve the manuscript. As the in vitro datasets were included as a technical control, the effect of other cellular components on the lattice were not considered directly relevant, and we simply needed a sample that robustly showed an expanded lattice as a control. We also consider any Taxol concentration dependence of the lattice spacing, either in cells or in vitro, beyond the scope of this work. The additional labels that were present on the tubulin were there to verify GMPCPP MT production with a light microscope. The reviewer is correct that these labels are not necessary for in vitro microtubule preparation. We therefore repeated our Taxol and GMPCPP in vitro experiments, without including the fluorophore and biotin labels (using fully unlabelled tubulin) and without a

freeze/thaw cycle of the prepared samples (see M&M lines **262-264** and **270-272**). This shows that without the labels, Taxol and GMPCPP-bound microtubules (N=10 for both, data now in supplemental figure 3F) are also expanded and that the conclusions drawn based on our original datasets still stand (**lines 95-97**).

The manuscript describes small changes in the microtubule lattice using precise image processing protocols. Therefore, these protocols need to be described in greater detail than what is provided now. For example, the authors need to be more detailed in the Methods section when describing how they performed subtomogram averaging to estimate protofilament number in microtubules and to determine helical symmetry parameters. What volume did they use as an initial reference? If the initial reference was a 13-stranded microtubule, how did they avoid reference bias? The authors should reconstruct individual microtubules (e.g., using approaches described in Foster, H.E. et al., 2022, JCB) to double-check protofilament number distribution in Taxol-treated and control cells. Additionally, how did the authors perform helical averaging to find helical parameters? Did they obtain initial helical parameters from a C1 reconstruction and then refined those parameters iteratively to obtain the final reported values?

Additional explanations for other parts of the workflow noted in response to reviewer 1.

We agree with the reviewer that the details for the subtomogram averaging workflow could be improved. We have now described the workflow in more detail (**lines 432-443**) and have clarified that the initial 3D refinement was performed with a round tubular reference to prevent bias. This refinement allowed us to determine the average protofilament number. We also clarify that we obtained the helical parameters by performing 3D classification using a 40 Å low pass filtered 13 PF reference and a broad search range of rise and twist values centred around values which are standard for a 13 PF microtubule. To clarify for the reader that the current resolution does not necessarily allow for determination of exact helical values, we note in the text that the rise is approximated (and not exactly determined) (**line 118-120**).

To determine the PF number of individual microtubules, using methods similar to Foster *et al.* JCB 2022, we reconstructed aligned particles originating from a single microtubule to obtain an average based on one microtubule. Then we took the sum of the average (64 slices, 8.68 Å/pixel, ~56 nm thickness) in the direction of the microtubule axis to create a projection. To this projection, we applied C11 to C16 symmetry to strengthen the signal and test 11PF to 16 PF numbers. This was repeated for 5 randomly selected microtubules. For all microtubules the C13 symmetry resulted in the most well defined protofilaments, indicating that they are 13 PF microtubules (Figure 1R). This is consistent with the Relion classification results.

Figure 1R: C11 to C16 symmetry operations were applied to the summed slice of a individual reconstructed microtubule (8.68 Å/pixel, boxsize 64 pixels, ~56 nm thickness). This was done for 5 individual microtubules. C13 symmetry yielded the best defined protofilaments in all examples.

Furthermore, given these are not *in vitro* prepared microtubules, we expect that the PF distribution will be much narrower in the cell where microtubule templating and regulatory proteins are present. From our data it appears that U2OS cells have a uniform 13 PF distribution. To our knowledge, an in cell protofilament number different than 13 has not been reported for human cells.

Fig.4 Panel J: The authors have tomography data collected on lamella areas with low or no StableMARK signal (Fig. S6). Microtubule lattice spacing distribution on those areas should be added to the graph.

We measured the lattice spacing of 12 microtubules which did not appear labelled by StableMARK. This yielded a distribution different from the StableMARK distribution, with compacted lattices occurring more frequently (Supplemental Figure 7F), consistent with our other data and our expectations. However, due to the limited performance of our cryo-fluorescent microscopes (air objectives, higher photon losses in the optical path, limited camera DQE, etc.), the system is not sensitive enough to detect faintly labelled microtubules which may also be present. Consequently, we do not want to emphasize to readers the use of our workflow to **exclude** the presence of certain factors. We also make this point in **lines 183-188** of the paper when mentioning our uncorrelated example.

Supplemental Figure 7F: Uncorrelated dataset (N=12, 10 tomograms, 8 cells) placed in between the StableMARK and untreated dataset for comparison.

A summary Supplementary Table listing cryo-electron tomography datasets and collection parameters for each of the datasets is needed.

We thank the reviewer for this useful suggestion and have added a supplementary table with all the parameters used (supplementary table 2).

September 12, 2024

RE: JCB Manuscript #202206143RR

Dr. Stuart Howes
Utrecht University
David de Wiedgebouw
Universiteitsweg 99
Utrecht, Utrecht 3584 CG
Netherlands

Dear Dr. Howes,

Thank you for submitting your revised manuscript entitled "StableMARK-decorated microtubules in cells have expanded lattices." We would be happy to publish your paper in JCB pending final revisions necessary to meet our formatting guidelines (see details below).

A. MANUSCRIPT ORGANIZATION AND FORMATTING:

1) Text limits: Character count for Reports is < 40,000, not including spaces. Count includes title page, abstract, introduction, results & discussion, and acknowledgments. Count does not include materials and methods, figure legends, references, tables, or supplemental legends.

2) Figure formatting: Reports may have up to 5 main text figures. Scale bars must be present on all microscopy images, including inset magnifications. Please avoid pairing red and green for images and graphs to ensure legibility for color-blind readers. If red and green are paired for images, please ensure that the particular red and green hues used in micrographs are distinctive with any of the colorblind types. If not, please modify colors accordingly or provide separate images of the individual channels.

3) Statistical analysis: Error bars on graphic representations of numerical data must be clearly described in the figure legend. The number of independent data points (n) represented in a graph must be indicated in the legend. Please, indicate whether 'n' refers to technical or biological replicates (i.e. number of analyzed cells, samples or animals, number of independent experiments). If independent experiments with multiple biological replicates have been performed, we recommend using distribution-reproducibility SuperPlots (please see Lord et al., JCB 2020) to better display the distribution of the entire dataset, and report statistics (such as means, error bars, and P values) that address the reproducibility of the findings.

Statistical methods should be explained in full in the materials and methods. For figures presenting pooled data the statistical measure should be defined in the figure legends. Please also be sure to indicate the statistical tests used in each of your experiments (both in the figure legend itself and in a separate methods section) as well as the parameters of the test (for example, if you ran a t-test, please indicate if it was one- or two-sided, etc.). Also, if you used parametric tests, please indicate if the data distribution was tested for normality (and if so, how). If not, you must state something to the effect that "Data distribution was assumed to be normal but this was not formally tested."

4) Materials and methods: Should be comprehensive and not simply reference a previous publication for details on how an experiment was performed. Please provide full descriptions (at least in brief) in the text for readers who may not have access to referenced manuscripts. The text should not refer to methods "...as previously described."

5) For all cell lines, vectors, constructs/cDNAs, etc. - all genetic material: please include database / vendor ID (e.g., Addgene, ATCC, etc.) or if unavailable, please briefly describe their basic genetic features, even if described in other published work or gifted to you by other investigators (and provide references where appropriate). Please be sure to provide the sequences for all of your oligos: primers, si/shRNA, RNAi, gRNAs, etc. in the materials and methods. You must also indicate in the methods the source, species, and catalog numbers/vendor identifiers (where appropriate) for all of your antibodies, including secondary.

6) Microscope image acquisition: The following information must be provided about the acquisition and processing of images:
a. Make and model of microscope
b. Type, magnification, and numerical aperture of the objective lenses

- c. Temperature
- d. Imaging medium
- e. Fluorochromes
- f. Camera make and model
- g. Acquisition software
- h. Any software used for image processing subsequent to data acquisition. Please include details and types of operations involved (e.g., type of deconvolution, 3D reconstitutions, surface or volume rendering, gamma adjustments, etc.).

7) References: There is no limit to the number of references cited in a manuscript. References should be cited parenthetically in the text by author and year of publication. Abbreviate the names of journals according to PubMed.

8) Supplemental materials: Reports generally have up to 3 supplemental figures and 10 videos. You currently exceed this limit but, in this case, we will be able to give you the extra space but if possible, please try to consolidate the supplemental figures.

9) eTOC summary: A ~40-50 word summary that describes the context and significance of the findings for a general readership should be included on the title page. The statement should be written in the present tense and refer to the work in the third person. It should begin with "First author name(s) et al..." to match our preferred style.

10) Conflict of interest statement: JCB requires inclusion of a statement in the acknowledgements regarding competing financial interests. If no competing financial interests exist, please include the following statement: "The authors declare no competing financial interests." If competing interests are declared, please follow your statement of these competing interests with the following statement: "The authors declare no further competing financial interests."

11) A separate author contribution section is required following the Acknowledgments in all research manuscripts. All authors should be mentioned and designated by their first and middle initials and full surnames. We encourage use of the CRediT nomenclature (<https://casrai.org/credit/>).

12) ORCID IDs: ORCID IDs are unique identifiers allowing researchers to create a record of their various scholarly contributions in a single place. Please note that ORCID IDs are required for all authors. At resubmission of your final files, please be sure to provide your ORCID ID and those of all co-authors.

13) Journal of Cell Biology now requires a data availability statement for all research article submissions. These statements will be published in the article directly above the Acknowledgments. The statement should address all data underlying the research presented in the manuscript. Please visit the JCB instructions for authors for guidelines and examples of statements at (<https://rupress.org/jcb/pages/editorial-policies#data-availability-statement>).

B. FINAL FILES:

Thank you for your attention to these final processing requirements. Please revise and format the manuscript and upload materials within 7 days. If you need an extension for whatever reason, please let us know and we can work with you to determine a suitable revision period.

Thank you for this interesting contribution, we look forward to publishing your paper in Journal of Cell Biology.

Sincerely,

Arshad Desai, PhD
Monitoring Editor
Journal of Cell Biology

Dan Simon, PhD
Scientific Editor
Journal of Cell Biology